



# Sentinel-1 snow depth retrieval at sub-kilometer resolution over the European Alps

Hans Lievens[1], Isis Brangers[1], Hans-Peter Marshall[2], Tobias Jonas[3], Marc Olefs[4], and Gabriëlle De Lannoy[1]

[1]Department of Earth and Environmental Sciences, KU Leuven, Leuven, Belgium
[2]Department of Geosciences, Boise State University, Boise, ID, USA
[3]WSL - Institute for Snow and Avalanche Research SLF, Davos, Switzerland
[4]ZAMG - Zentralanstalt für Meteorologie und Geodynamik, Vienna, Austria

**Correspondence:** Hans Lievens (Hans.Lievens@KULeuven.be)

**Abstract.** Seasonal snow in mountain regions is an essential water resource. However, the spatio-temporal variability in mountain snow depth or snow water equivalent (SWE) from regional to global scales is not well understood due to the lack of high-resolution satellite observations and robust retrieval algorithms. We demonstrate the ability of the Sentinel-1 mission to monitor weekly snow depth at sub-kilometer (100 m, 300 m and 1 km) resolutions over the European Alps, for 2017–2019. Sentinel-1 backscatter observations, especially for the cross-polarization channel, show a high correlation with regional model simulations of snow depth over Austria and Switzerland. The observed changes in radar backscatter with the accumulation or ablation of snow are used in a change detection algorithm to retrieve snow depth. The algorithm includes the detection of dry and wet snow conditions. For dry snow conditions, the 1 km Sentinel-1 retrievals have a spatio-temporal correlation (R) of 0.87 and mean absolute error (MAE) of 0.17 m compared to in situ measurements across 743 sites in the European Alps. A slight reduction in performance is observed for the retrievals at 300 m (R = 0.85 and MAE = 0.18 m) and 100 m (R = 0.79 and MAE = 0.21 m). The results demonstrate the ability of Sentinel-1 to provide regional snow estimates at an unprecedented resolution in mountainous regions, where satellite-based estimates of snow mass are currently lacking. The retrievals can improve our knowledge of seasonal snow mass in areas with complex topography and benefit a number of applications, such as water resources management, flood forecasting and numerical weather prediction.

## 1 Introduction

In the European Alps, the release of precipitated water to discharge is delayed by storage in snow and glaciers. During the spring and summer, when water demand is high, snow and glacier meltwater sustains the more than 14 million inhabitants across 8 countries within the Alpine region, e.g., by supplying water for domestic use, industry, hydropower production, agriculture, etc. However, climate change causes mass loss of most glaciers (Zemp et al., 2019) and perturbs snowmelt dynamics (Bor-





mann et al., 2018; Pulliainen et al., 2020), changing the timing and magnitude of water availability (Immerzeel et al., 2020). An improved monitoring of snow water resources can help strengthening our understanding of these changing hydrological processes in mountain regions.

Remote sensing can play an essential role in the monitoring of snow water resources. The current operational satellite re-
trievals of snow depth (or snow water equivalent, SWE, using auxiliary snow density information) rely primarily on passive microwave observations, either based only on remote sensing data (Kelly et al., 2003), or in combination with in situ measurements as in GLOBSNOW (Takala et al., 2011). Unfortunately, passive microwave observations have shortcomings, especially in mountain areas. The coarse footprints ($\sim$25 km) cannot resolve the high spatial variability in snow depth imposed by complex topography (Dozier et al., 2016), and the microwave observations have a tendency to saturate at $\sim$0.8 m snow depth, which
is often exceeded in mountains (Foster et al., 2005; Tedesco and Narvekar, 2010). Furthermore, missing details about the snow microstructure and layering complicate the physically-based retrievals (Lemmetyinen et al., 2018). Therefore, new and robust satellite observations are critically needed to fill the mountain-snow observation gap (Bormann et al., 2018).

Active microwave observations from Synthetic Aperture Radar (SAR) show promise for mapping snow depth or SWE at a high spatial resolution. The optimal frequencies for global-scale snow observations are likely within X- to Ku-band ($\sim$8–
18 GHz) (Rott et al., 2010), based on their strong scattering within the snow volume (Yueh et al., 2009; King et al., 2015). Unfortunately, high-resolution Ku-band satellite observations are not available at present. Alternatively, the use of SAR interferometry (InSAR) (Guneriussen et al., 2001; Leinss et al., 2015; Conde et al., 2019) allows for tracking changes in SWE from changes in the radar signal phase that are caused by refractions in the snowpack. For this approach, lower frequencies such as L-band (1-2 GHz) are preferred, maintaining a better coherence between repeat observations. A number of future missions
are addressing the use of L-band InSAR for snow applications, such as the National Aeronautics and Space Administration (NASA) - Indian Space Research Organisation (ISRO) SAR (NISAR) and potentially the Radar Observing System for Europe - L-Band (Rose-L).

Currently, routine SAR backscatter ($\sigma^0$) observations with a high spatial resolution ($\sim$20 m) and frequent revisit ($<$weekly) are only available at C-band (5.4 GHz) from the European Space Agency (ESA) and Copernicus Sentinel-1 (S1) mission.
Despite the availability of these routine observations, limited attention has been drawn to the use of C-band $\sigma^0$ for snow monitoring, after earlier satellite observations had shown a limited sensitivity (Rott and Nagler, 1993; Bernier and Fortin, 1998; Bernier et al., 1999; Shi and Dozier, 2000). However, these studies were mostly focused on relatively shallow snow depths (below $\sim$1 m) and were investigating the use of $\sigma^0$ observations in co-polarization. At C-band, co-polarized $\sigma^0$ is generally dominated by (i) the scattering from the ground surface (depending on the moisture content and the temperature
or freeze-thaw state of the soil) during dry snow conditions, or by (ii) the absorption by liquid water in wet snow conditions (Baghdadi et al., 1997; Nagler and Roth, 2000; Luojus et al., 2007; Nagler et al., 2016).

Observations at C-band in cross-polarization are in principle more sensitive to dry snow accumulation than in co-polarization for two reasons. First, the surface scattering from the ground is significantly weaker in cross-polarization. Second, dry snow represents a dense medium of irregularly-shaped and clustered ice crystals that primarily causes volume scattering in cross-
polarization (Chang et al., 2014). Hence, the surface scattering from the ground may no longer dominate over the volume





scattering from the snow (Shi and Dozier, 2000; Pivot, 2012), especially for deep snow that is often encountered in mountain regions. Recently, Lievens et al. (2019) demonstrated the sensitivity of S1 cross-polarized $\sigma^0$ observations to dry snow accumulation and developed an empirical change detection algorithm to retrieve snow depth at 1 km spatial resolution over all mountain ranges in the Northern Hemisphere. The retrievals are based on a number of assumptions that, in the case of C-band,

are likely more valid for deeper snow. That is, (i) an increase in snow depth causes an increase in snow volume scattering and thus also in cross-polarized $\sigma^0$, (ii) the snow volume scattering contribution is not negligible compared to the ground surface scattering contribution, and (iii) the ground surface scattering contribution remains relatively constant during the winter period because of the limited changes in soil temperature (due to the insulating properties of snow), soil moisture and surface roughness, such that (iv) the main changes in $\sigma^0$ over time can be related to changes in snow depth.

Here, we further refine the S1 snow depth retrievals and evaluate them at 100 m, 300 m and 1 km over the European Alps for the two year period from August 2017 through July 2019. First, The sensitivity of the S1 $\sigma^0$ observations in co- and cross-polarization is evaluated using regional model simulations of snow depth over Austria and Switzerland and stratified by elevation and forest cover fraction. Thereafter, S1 snow depth retrievals at three resolutions are regionally optimized using auxiliary information on topography and land cover. Finally, the accuracy of the retrievals is estimated based on independent

time series measurements at 743 in situ locations across the Alps.

The results of this study contribute to improving our knowledge of seasonal snow mass in areas with complex topography and thereby addresses a long-standing observation gap in the remote sensing of the cryosphere. A strong asset is the assured long-term continuity of S1 C-band SAR observations over the coming decades, which will allow for analyzing trends in snow mass impacted by climate variability or climate change. Finally, the S1 snow depth retrievals could be of high value for data

assimilation into land surface models (Girotto et al., 2020). Not only could the assimilation ensure improved and continuous (in time and space) estimates of various snow variables, it is likely to also benefit applications such as flood forecasting (Dechant and Moradkhani, 2011; Griessinger et al., 2019) or numerical weather prediction (de Rosnay et al., 2014).

## 2 Data and methods

### 2.1 Sentinel-1 observations

The S1 mission is a constellation of two satellites, S1A and S1B, launched in April 2014 and 2016, respectively. Over land and outside the polar regions, S1 routinely operates in the Interferometric Wide Swath (IW) mode, acquiring $\sigma^0$ observations at ~20 m resolution in vertical-vertical (VV) and vertical-horizontal (VH) polarizations. We processed the IW S1A and S1B ground-range detected (GRD) $\sigma^0$ data for the period August 1, 2017 to July 31, 2019, over the European Alps (~4600 images). The processing was performed using the ESA Sentinel Application Platform (SNAP) toolbox, applying standard processing

techniques: precise orbit file application, GRD border noise removal, thermal noise removal, radiometric calibration, multi-looking to 100 m resolution and range-Doppler terrain correction with projection onto the World Geodetic System WGS84 geopgraphic coordinate system. The 100 m grid cell size was chosen to reduce the speckle noise inherent to radar observations





and to limit the impacts of geometric distortions (e.g., radar foreshortening, layover and shadowing) that are enhanced by the topographic complexity of the Alps.

The S1A and S1B satellites have an exact 12-day repeat cycle, with 175 orbits per cycle. As both satellites share the same orbital plane with a 6-day offset, the two-satellite constellation offers an exact 6-day repeat cycle. Depending on the

region, more frequent observations (2–3 days on average) are available due to orbits with partly overlapping swaths and the combination of ascending and descending tracks.

For a given 100 m grid cell, the S1 observations from different orbits within one repeat cycle have a different viewing geometry, in terms of look direction, incidence angle and azimuth angle. Particularly in mountainous areas, the impact of these different viewing geometries on $\sigma^0$ can be large, for instance when a terrain slope is facing towards or away from the sensor

line of sight, and thus needs to be accounted for. To reduce these effects, we corrected $\sigma^0$ for the local incidence angle ($\theta$; in degrees), i.e., the angle between the radar line of sight and the normal to the local surface. The local surface was derived from the Shuttle Radar Topography Mission (SRTM) digital elevation model (projected onto the 100 m WGS84 grid). The $\sigma^\circ$ correction to the 40° reference incidence angle was based on the frequently applied cosine approximation (Ulaby et al., 1982; Bernier and Fortin, 1998):

$$\sigma^0_{40^\circ} = \sigma^0 \frac{\cos 40^\circ}{\cos \theta} \tag{1}$$

Future research is recommended to investigate the use of observations processed to terrain-flattened backscatter ($\gamma^0$), which offers an improved correction for the local incidence angle by also taking into account its impact on the illuminated area (Small, 2011; Small et al., 2021). Further, we applied a bias correction between the observations from the different orbits. Over repeating cycles, acquisitions from a given orbit consistently observe the same grid cell with the same incidence and azimuth

angle, and should thus provide unbiased repeated samples. Therefore, we pooled the $\sigma^0$ values pertaining to the same orbits into separate time series. For each orbit, we calculated the first and second order moment statistics (temporal mean and standard deviation), and scaled the $\sigma^0$ values (in dB) such that these statistics match the average statistics across all orbits:

$$\sigma^0_{40^\circ,o} = <\overline{\sigma^0_{40^\circ}}> + \left(\sigma^0_{40^\circ,o} - \overline{\sigma^0_{40^\circ,o}}\right) \frac{<\text{std}\left(\sigma^0_{40^\circ}\right)>}{\text{std}\left(\sigma^0_{40^\circ,o}\right)} \tag{2}$$

with $\sigma^0_{40^\circ,o}$ the bias-corrected backscatter for orbit $o$, $\overline{\sigma^0_{40^\circ,o}}$ the temporal mean backscatter for orbit $o$, $\text{std}\left(\sigma^0_{40^\circ,o}\right)$ the temporal

standard deviation in backscatter for orbit $o$, and $<>$ denoting the average over the different orbits. This procedure ensures unbiased $\sigma^0$ values in the mean and standard deviation across orbits. Note that for the calculation of the time series statistics (mean and standard deviation), we excluded observations from March through July, to avoid strong $\sigma^0$ fluctuations caused by wet snow, with potentially large differences between 6 am descending and 6 pm ascending tracks in case of refreezing.

As the final processing steps, we performed an outlier correction, by replacing values that differ more than $\pm 1.5$ dB from

the mean $\sigma^0$ calculated over a 12-day window by the mean value. The 12-day window was selected as this corresponds with the S1 repeat cycle and thus ensures the inclusion of observations obtained from the same platform (S1A or S1B) and orbit. Subsequently, we averaged the available $\sigma^0$ observations to weekly values. The result of the processing is thus weekly 100 m



(co- and cross-polarization) backscatter at 40° over the Alps, orbit-bias corrected, and obtained by merging ascending and descending S1A and S1B observations. In the remainder of the text, we denote the final processed backscatter values as $\sigma^0$ for simplicity.

## 2.2 Snow depth retrievals

The snow depth retrieval algorithm is based on the reasoning that snow is a dense medium of clustered, irregularly shaped, ice crystals that contribute to volume scattering (Chang et al., 2014). A deeper snowpack in theory results in stronger volume scattering, and therefore, the strength of the volume scattering can be related with snow depth. While higher microwave frequencies (such as X- and Ku-band) are likely more suitable to detect volume scattering in shallow snow environments, we hypothesize that the contribution at C-band is large enough for application to the typically deep snow regimes in mountain

areas. Snow volume scattering generally has a stronger contribution in cross-polarized (VH in the case of S1) compared to co-polarized (VV) observations, whereas scattering impacts from soil moisture, soil freeze/thaw or soil temperature are expected to be more similar in co- and cross-polarization. Therefore, the use of a ratio (in linear scale, or difference in dB scale) between cross- and co-polarization $\sigma^0$ observations can reduce the impacts of ground, vegetation and surface geometry properties, and enhance the sensitivity to snow depth (Bernier et al., 1999; Lievens et al., 2019).

Although based on the physical principle of an increase in snow volume scattering with increase in snow depth, the retrieval algorithm uses an empirical change detection approach. The method builds further on Lievens et al. (2019) and the main differences with this study are highlighted in this section. As auxiliary input, the algorithm only requires snow cover (SC) absence/presence observations, which we derived from the 1 km resolution Interactive Multisensor Snow and Ice Mapping System (IMS; National Ice Center (2008)) dataset, and forest cover fraction (FCF), derived from the 100 m Copernicus PROBA-

V dataset (Buchhorn et al., 2020).

Firstly, the change detection algorithm from Lievens et al. (2019) has been modified to account for the observation that, for sparsely vegetated areas, a ratio of VH- and VV-polarized $\sigma^0$ (cross-polarization ratio; CR) shows the highest correlation with snow depth, whereas in more densely forested areas, the VV-polarized $\sigma^0$ shows a higher temporal correlation (see Fig. 5 below). The latter may be caused by the relatively stronger scattering contribution from forests (and vegetation in general)

in VH-polarization (Vreugdenhil et al., 2020), decreasing the sensitivity to the underlying snowpack. Therefore, two change detection indices (SI1 and SI2; in dB) are calculated for each location $i$ and weekly time step $t$:

$$\text{SI1}(i,t) = \text{SI1}(i,t-1) + \left[\sigma^0_{\text{CR}}(i,t) - \sigma^\circ_{\text{CR}}(i,t-1)\right] \tag{3}$$

$$\text{SI2}(i,t) = \text{SI2}(i,t-1) + \left[\sigma^0_{\text{VV}}(i,t) - \sigma^\circ_{\text{VV}}(i,t-1)\right] \tag{4}$$

where $\sigma^0_{\text{CR}}$ represents a weighted combination of cross-polarized $\sigma^0_{\text{VH}}$ and co-polarized $\sigma^0_{\text{VV}}$ observations (in dB) of the general

form:

$$\sigma^0_{\text{CR}} = A\sigma^0_{\text{VH}} - \sigma^0_{\text{VV}} \tag{5}$$





with $A$ being a fitting parameter. SI1 and SI2 are set to zero when snow cover (SC) is absent, or in case of negative values. Strong temporal changes in SI (above 0.5 dB) are reduced (by a factor of 0.3) to avoid outliers. Further, SI1 and SI2 are scaled and combined based on the PROBA-V forest cover fraction (FCF (-); ranging from 0 to 1):

$$\mathrm{SI} = (1 - \mathrm{FCF}) \cdot \frac{1}{1 - B \cdot \mathrm{FCF}} \cdot \mathrm{SI1} + \mathrm{FCF} \cdot C \cdot \mathrm{SI2} \tag{6}$$

with $B$ and $C$ fitting parameters. The three algorithm parameters were iterated over $A \in [1 : 1 : 3]$, $B \in [0 : 0.1 : 1]$ and $C \in [0 : 0.1 : 2]$ and optimized based on model simulations of snow depth (optimization not shown; simulations described in Section 3.1). This resulted in $A = 2$, $B = 0.7$ and $C = 1.4$, which are further used in this study. Note that the optimized value for $B$ is similar to the value of 0.6 used in Lievens et al. (2019), which is also used in the operational passive microwave retrievals (Tedesco and Narvekar, 2010).

Secondly, as an extension to Lievens et al. (2019), we here include a wet snow detection algorithm to mask out the S1 retrievals in wet snow conditions. Wet snow is known to absorb a large part of the radar signal, causing a strong decrease in $\sigma^0$ (e.g., Baghdadi et al., 1997; Nagler and Roth, 2000; Luojus et al., 2007; Nagler et al., 2016; Marin et al., 2020) that could result in an underestimation in snow depth. For each weekly time step $t$, the wet snow detection criterion applied is:

$$\sigma_{\mathrm{VH}}^0(t+1) - \sigma_{\mathrm{VH}}^0(t-1) < D \quad \text{or} \quad \sigma_{\mathrm{VV}}^0(t+1) - \sigma_{\mathrm{VV}}^0(t-1) < D \tag{7}$$

with $D$ a threshold set to -1.25 dB. Note that we refrain from detecting wet snow based on the difference in $\sigma^0$ with a (constant) dry snow reference value, as often done in the literature (e.g., Baghdadi et al., 1997; Nagler et al., 2016; Tsai et al., 2019; Manickam and Barros, 2020), because of the difficulty to accurately define the dry snow reference value (Manickam and Barros, 2020). For instance, an average summer $\sigma^0$ is often used as an approximation of the dry snow reference (Tsai et al., 2019; Manickam and Barros, 2020). However, in regions with dense vegetation, summer $\sigma^0$ can be high due to vegetation 20   contributions, causing an over-detection of wet snow in winter if $\sigma^0$ is systematically lower. Conversely, in bare or sparsely vegetated regions, snow accumulation can cause a significantly higher $\sigma^0$ (by several dB) compared to the summer reference, requiring a large decrease from wet snow to reach the threshold below the summer reference (see below, e.g., Fig. 4). The use of a fixed dry snow reference value is here avoided by thresholding the decrease rate of $\sigma^0$ over time instead. After wet snow is detected, the retrievals are masked over subsequent time steps until snow cover is absent (but can then be re-initiated 25   if snow becomes present again). To avoid the masking of complete time series due to early (autumn) wet snow, the detection is only initiated in February. Furthermore, $\sigma^0$ observations that are 3 dB below the September–November mean (in either VV- or HV-polarization) are excluded to reduce the impacts of decreases in $\sigma^0$ by early wet snow, and also by soil freezing (Baghdadi et al., 2018).

Thirdly, we translate the SI (in dB) into snow depth (or height of snow, HS, in m) by multiplication with an empirical scaling 30   parameter $E$:

$$\mathrm{HS}(i,t) = E \cdot \mathrm{SI}(i,t) \tag{8}$$

The value of $E$ is estimated by minimizing the absolute bias between the retrieved and the modeled snow depth (Section 3.1). Note that the scaling based on model simulations can propagate uncertainties in the simulations to the retrieval. Two estimates





for $E$ are proposed, i.e. $E_1$ and $E_2$. As a first approximation, $E_1$ is considered constant in space and time (Lievens et al., 2019). The $E_1$ value that minimizes the bias against the model simulations equals 0.56 m/dB. Thus, a 1 dB temporal change in the input $\sigma^0$ ratio on average corresponds with a 0.56 m change in snow depth. Besides a constant scaling factor, we also investigate a more optimized estimation of $E_2$ that varies in space, depending on auxiliary information of topography (SRTM

elevation) and land cover (Copernicus PROBA-V classification). Therefore, $E_2$ is obtained by first stratifying the optimization by elevation (i.e., selecting the value that minimizes the bias per elevation band, ranging from 50 m to 3500 m in steps of 100 m), and subsequently stratifying the optimization by land cover class.

Using the algorithm outlined above, we processed snow depth retrievals for the 2 year period from August 1, 2017 to July 31, 2019 over the entire European Alps. The retrievals are performed at three different spatial resolutions, i.e., 100 m, 300 m and

1 km, to assess the impact of the spatial resolution on the retrieval performance. The 300 m and 1 km retrievals are obtained by linearly averaging the retrievals at 100 m. Note that the retrievals at the finer scales (i.e., 100 m, 300 m) use the same relationship of $E_2$ with elevation and land cover, as established based on the 1 km data (because the snow depth model simulations used for the optimization are only available at the 1 km scale).

## 2.3 Model simulations

Output from two regional snow models for Austria (SNOWGRID-CL) and Switzerland (OSHD) was used to assess S1 backscatter and snow depth retrievals. Fig. 1 shows that the joint simulation domain covers a large part of the Alps. For the comparison with the weekly 1 km S1 observations, the model simulations of snow depth from the SNOWGRID-CL and OSHD were averaged from the daily to the weekly time scale. In addition, simulations by the Noah-Multiparameterization (Noah-MP) land surface model (Niu et al., 2011) were used to assess the impacts of soil moisture and soil temperature on S1

backscatter.

SNOWGRID-CL (Olefs et al., 2020) is the climate version of the spatially distributed snow model ran operationally at ZAMG with a daily timestep on a 1 km grid over Austria. The model outputs snow depth and SWE, and is forced with daily 1 km gridded meteorological data of air temperature, precipitation and evaporation that take into account the high variability of these variables in complex terrain (Hiebl and Frei, 2016, 2018; Haslinger and Bartsch, 2016). The model solves the shortwave

radiation balance following Pellicciotti et al. (2005) and uses a simple 2-layer snow scheme, considering settling, the exchange of heat and liquid water content, and energy addition by rain. Lateral snow mass redistribution is included as a function of snow and terrain properties (Frey and Holzmann, 2015). The accuracy of the daily snow depth simulations (correlation of 0.83, root-mean-square error (RMSE) of 14.11 cm, and bias of -3.12 cm) has been assessed for the period from November through April 2011–2018 (Olefs et al., 2020).

The OSHD (operational snow-hydrological service) multi-model framework provides daily SWE and snowmelt estimates for Switzerland at 1 km spatial resolution. It consists of a suite of spatially distributed snow models integrated with three-dimensional sequential assimilation of snow monitoring data from several hundred sites (Magnusson et al., 2014; Winstral et al., 2019). The models include procedures to account for unresolved variability at the sub-grid scale, two of which are: (1) fractional snow-covered area is parameterized using terrain parameters derived at 25 m spatial resolution according to Helbig



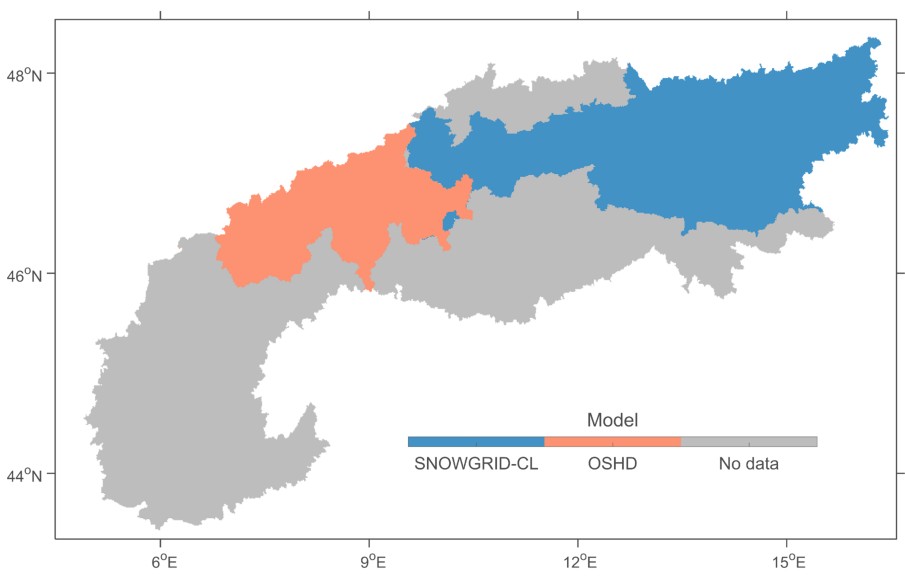

**Figure 1.** The domains of the 1 km resolution SNOWGRID-CL and OSHD model simulations of snow depth.

et al. (2015); (2) snow distribution and redistribution at small scales is considered using slope- and aspect-dependent transfer functions that were derived from a set of high-resolution snow depth maps from airborne lidar (Grünewald and Lehning, 2015). For this study we used output from the JULES Investigation Model (JIM; Essery et al. (2013)), which is one of the OSHD members. JIM was forced using a combination of numerical weather prediction output from COSMO-1 (www.cosmo-model.org) and reanalysis data as detailed in Winstral et al. (2019), and includes data assimilation from 344 Swiss snow monitoring stations. The assimilation improves the accuracy of the daily snow depth simulations from an RMSE of 21.3-42.4 cm and bias of 4.08-26.1 cm (varies between the years 2014, 2015 and 2017) to an RMSE of 17.3-25.6 cm and bias of 0.8-2.5 cm.

Additional simulations with the Noah-MP land surface model are performed to provide information on soil conditions (SNOWGRID-CL and OSHD are strictly snowpack models). These simulations cover the entire Alps at the daily timescale and are forced using the globally-available Modern-Era Retrospective analysis for Research and Applications, Version 2 dataset (Gelaro et al., 2017). The analyzed model outputs include soil moisture (in $m^3 \, m^{-3}$) for the top soil layer (0–0.1 m) and soil temperature (in K) for four soil layers (0–0.1 m, 0.1–04 m, 0.4–1 m and 1–2 m).

## 2.4 In situ measurements

Weekly average in situ measurements of snow depth for the period August 1, 2017 to July 31, 2019 were assembled for 743 sites, managed by Météo France, WSL–SLF and ZAMG. The locations of the sites are shown in Fig. 2 on top of SRTM elevation and PROBA-V land cover. The minimum, mean and maximum elevations of the measurement sites are 230 m, 1395 m and 3114 m, respectively. The in situ time series were used to evaluate the temporal variability of S1 retrievals in terms of

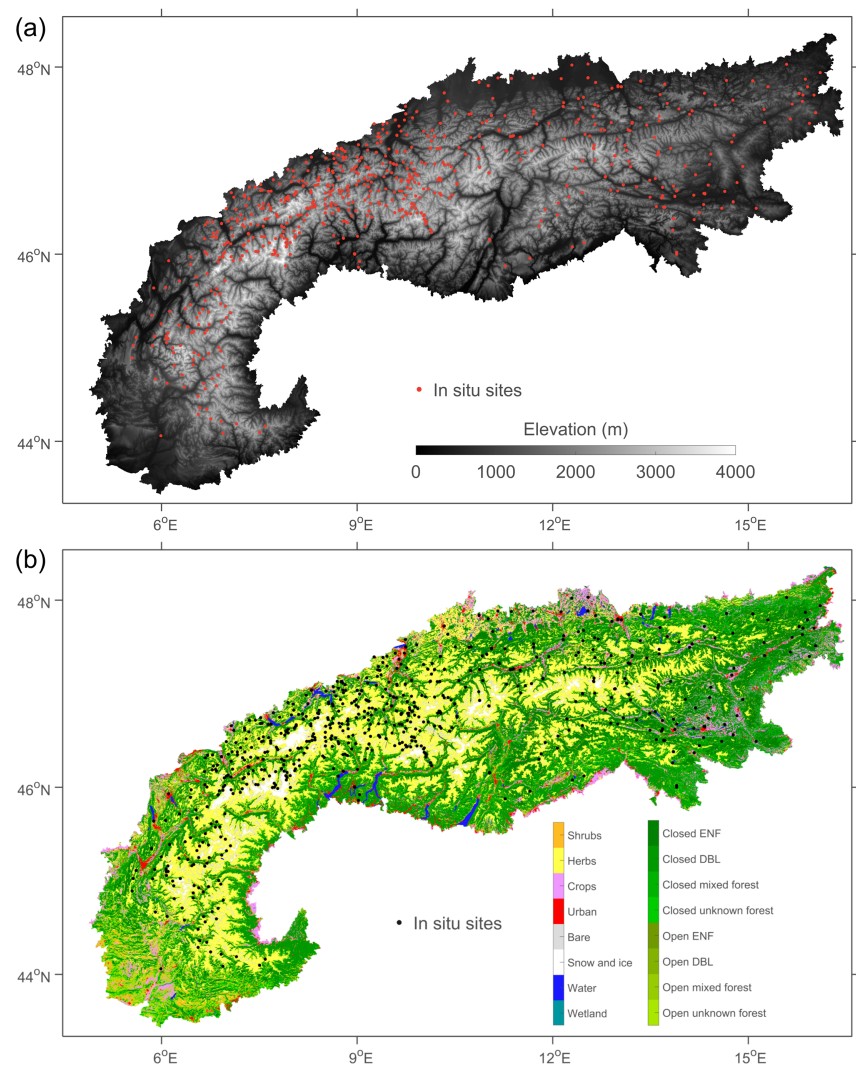

**Figure 2.** The in situ measurement locations within the European Alps on top of (a) elevation (m) and (b) land cover class (-).

Pearson correlation (R), bias and mean absolute error (MAE), and the large spatial coverage of the sites is used to compute spatial R, bias and MAE at the weekly time scale during the winter seasons.

An inevitable problem in the comparison between S1 observations and in situ measurements is that, particularly in mountain areas, the point-scale snow conditions at the in situ sites not always resemble the grid-scale conditions represented by the satellite data. The local variability in conditions can be large due to differences in elevation, slope and aspect, as well as wind and vegetation impacts on snow distribution (Blöschl, 1999; Seidel et al., 2016; Schattan et al., 2017). Moreover, in situ sites are preferentially located in relatively flat and non-forested terrain, often not representative of the surrounding area displaying





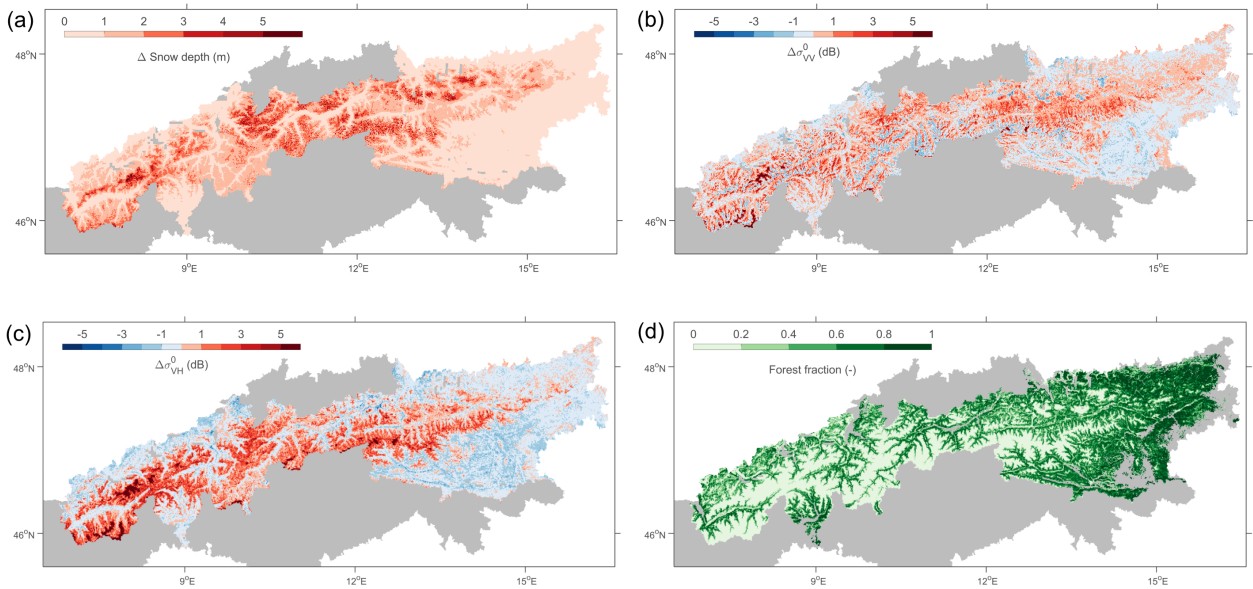

**Figure 3.** (a) The difference between the maximum weekly 1 km simulated snow depth (m) and the snow depth in the first week with snow cover for August–March (excluding the wet snow conditions detected by S1). (b) The corresponding observed difference in S1 $\sigma_{VV}^0$ (dB). (c) The corresponding difference in S1 $\sigma_{VH}^0$ (dB). (d) The PROBA-V forest cover fraction (-).

large variations in slope and forest cover (Meromy et al., 2013; Grünewald and Lehning, 2015; Dozier et al., 2016), and are underrepresented at high elevations (see Fig. 2).

## 3 Results and discussion

### 3.1 Comparison of 1 km S1 backscatter with snow model simulations

5 The response of S1 $\sigma^0$ in VV- and VH-polarization to dry snow accumulation is shown in Fig. 3. More specifically, the figure shows the simulated difference between the maximum (weekly) snow depth and the snow depth during the first week with snow cover during the period August–March, as well as the corresponding observed S1 differences in $\sigma^0$ between the weeks with the maximum snow depth and first snow cover (excluding wet snow conditions as detected by S1). The 1 km $\sigma^0$ was obtained by linearly averaging the values at the 100 m scale. The forest cover fraction from PROBA-V is also shown to support 10 the analysis.

In regions with shallow maximum snow depths ($<1$ m), $\sigma_{VV}^0$ generally remains relatively constant, with changes typically within $\pm 1$ dB (Fig. 3b). Most of the regions with significant snow accumulation, outside glaciated areas, show a slight increase in $\sigma_{VV}^0$ (about 1–3 dB). The underlying physical mechanisms that cause this increase are still uncertain, but we hypothesize that, in addition to snow volume scattering, the frequent occurrence of snow melt/refreeze cycles may lead to a more complex





snowpack with pronounced layering (incl. ice crusts and frozen percolation features) that increases scattering in co-polarization. The increase is mainly observed in areas with a lower $\sigma^0_{\mathrm{VV}}$ during snow-free conditions (not shown). In other regions, particularly at high elevations (above $\sim$2400 m, but not in glaciated areas), a slight decrease in $\sigma^0_{\mathrm{VV}}$ is observed. At these elevations, melt/refreeze cycles are much less frequent, causing a more homogeneous snowpack from which the scattering contributions

in co-polarization are lower compared to the contributions from the ground surface. The slight decrease in $\sigma^0_{\mathrm{VV}}$ in these regions may be caused by (a combination of) decreasing soil temperature (incl. soil freezing) and soil moisture, and also the attenuation of the ground surface scattering by the snowpack. Finally, in glaciated areas, a moderate to strong increase (4–5 dB or more) is observed, which is likely caused by the gradual freezing of glacial meltwater during winter, reducing the absorption of the radar signal through winter.

These results are in close agreement with several previous studies. Rott and Nagler (1993) found a slight decrease in European Remote Sensing (ERS) $\sigma^0_{\mathrm{VV}}$ observations over a shallow snowpack, but a strong increase over glaciated areas. Bernier and Fortin (1998) and Arslan et al. (2006) observed mostly an increase in co-polarized $\sigma^0$ during dry snow accumulation, based on RADARSAT and airborne observations, respectively. Shi and Dozier (2000) and Pivot (2012) observed an increase in RADARSAT and ERS co-polarized $\sigma^0$ (up to several dB) with dry snow accumulation in regions where snow volume scattering

was dominating over ground surface scattering, and a slight decrease in regions where the dominant effect was the attenuation of the ground surface scattering by the snowpack.

The signature in $\sigma^0_{\mathrm{VH}}$ (Fig. 3c) shows a more pronounced spatial pattern compared to $\sigma^0_{\mathrm{VV}}$, with a consistent and moderate to strong increase in $\sigma^0_{\mathrm{VH}}$ (about 3–4 dB) in regions where snow depth reaches above 1 m. We hypothesize that for cross-polarized observations the snow volume scattering is more consistently dominating over the (attenuated) ground surface scattering. The

ground scattering contribution is relatively lower, since it is mostly in the form of surface scattering with limited depolarization (especially for smooth and sparsely vegetated surfaces). The dry snow scattering contribution is stronger, because snow is a dense medium composed of a mixture of air and clustered, anisotropic ice crystals that cause volume scattering with depolarization. As in the $\sigma^0_{\mathrm{VV}}$ observations, the largest increases in $\sigma^0_{\mathrm{VH}}$ ($>$5 dB) are again observed over glaciated areas, and are likely caused by the combination of freezing glacial meltwater and snow accumulation.

Only a few previous studies have investigated satellite or airborne cross-polarized $\sigma^0$ observations at C-band in relation with snow depth or snow mass. Some of the heritage satellite missions, such as ERS, did not have the capability to measure in cross-polarization. Moreover, since the 90's, there has been a focus shift towards the use of higher-frequency microwave observations (e.g., in X- and Ku-band), in part due to the limited potential of C-band found with co-polarization, but also in preparation for higher-frequency satellite mission candidates (e.g., CoReH2O). Nevertheless, our results are in agreement with

Arslan et al. (2006), who observed a stronger increase in cross-polarization than in co-polarization at C-band with dry snow accumulation. More recently, Lievens et al. (2019) demonstrated the sensitivity of S1 cross-polarized $\sigma^0$ observations to snow depth over all Northern Hemisphere mountain ranges.

An exception to the overall higher correlation between simulated snow depth and S1 $\sigma^0_{\mathrm{VH}}$ is observed in regions with substantial snow accumulation and dense forest cover. Here, slightly higher correlations are found for $\sigma^0_{\mathrm{VV}}$. We hypothesize

this may be caused by the relatively stronger scattering from vegetation in cross-polarization (compared to co-polarization),



**Figure 4.** Time series of S1 $\sigma^0$ (dB) in (a,b) VH-polarization and (c,d) VV-polarization, compared against model simulations of snow depth (m) for a grid cell in (a,c) Austria (47.1341°N, 11.5928°E, 2355 m, simulations by SNOWGRID-CL) and (b,d) Switzerland (46.0921°N, 7.2539°E, 2140 m, simulations by OSHD). The $\sigma^0$ observations during snow-free conditions (derived from the model simulations) and wet snow conditions (derived from the $\sigma^0$ data) are displayed in grey. (e,f) show corresponding model simulations of soil moisture (SM1: 0–0.1 m; in m$^3$ m$^{-3}$) and soil temperature (T1: 0–0.1 m, T2: 0.1–04 m, T3: 0.4–1 m and T4: 1–2 m; in K) from the Noah-MP land surface model.

reducing the sensitivity to the scattering contributions from the snowpack. At the same time, forested regions in the Alps are mostly present at lower elevations in the valleys (below ~2500 m), where an increased occurrence of melt/freeze cycles could result in a more complex and layered snow structure that impacts the polarimetry.

Figure 4 shows time series examples of $\sigma^0$ in VH- and VV-polarization, along with model simulations of snow depth from 5   SNOWGRID-CL and OSHD, and soil moisture and soil temperature from Noah-MP, for two locations in Austria and Switzerland, respectively. The figure corroborates the results shown in Fig. 3. The $\sigma^0_{\text{VH}}$ observations show a strong correspondence

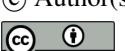



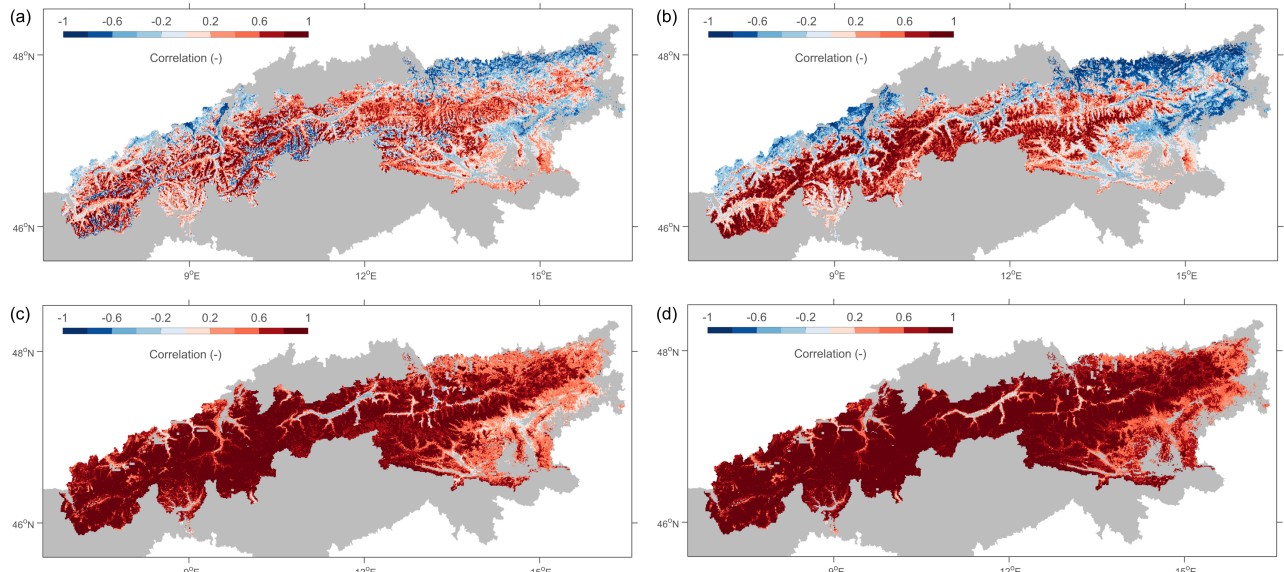

**Figure 5.** (a) The time series correlation between weekly simulated snow depth (m) and (a) S1 $\sigma_{VV}^0$ (dB), (b) $\sigma_{VH}^0$ (dB), (c) S1 snow depth (m), limited to the period with snow cover (based on the model simulations) and dry snow conditions (detected from S1), and (d) S1 snow depth (m), limited to the period with dry snow conditions but including zero snow depths.

with the dry snow accumulation at both locations and for both winter seasons. The evolution in $\sigma_{VH}^0$ is unlikely to be related to changes in soil moisture or soil temperature. Soil moisture shows a decrease throughout the snow accumulation period, and this would result, if it would have anywere of impact, in a decrease in $\sigma^0$. Soil temperature remains nearly constant throughout the snow accumulation period due to the insulating properties of snow. Near the peak of the snow season (although earlier in 2019 for the location in Austria), the strong decrease in $\sigma^0$ is caused by the increase in liquid water content within the snowpack that absorbs the radar signal. The wet snow detection algorithm effectively tracks the onset of the wet snow conditions, and allows for masking the subsequent observations (shown in grey). Note that the first wetting of the snow typically starts before the actual melting phase with shrinking of the snowpack (Marin et al., 2020). The $\sigma_{VV}^0$ observations correspond well with the snow depth simulations for the location in Switzerland, but not for the location in Austria, in particular during the winter of 2018–2019. There, the $\sigma_{VV}^0$ corresponds more closely with the soil moisture content, which suggests a more dominant scattering contribution from the ground surface.

## 3.2 Evaluation of 1 km S1 retrievals with snow model simulations

Figure 5 illustrates the time series correlations between the snow depth model simulations and the S1 $\sigma_{VV}^0$, S1 $\sigma_{VH}^0$ as well as the associated S1 snow depth retrievals, for the model domain at 1 km resolution covering Austria and Switzerland. The analysis for Figs. 5a–c has been limited to the periods from August to March 2017–2018 and 2018–2019, excluding time steps with zero snow depth (based on the model simulations) and time steps with wet snow (detected by S1). Consistent with the





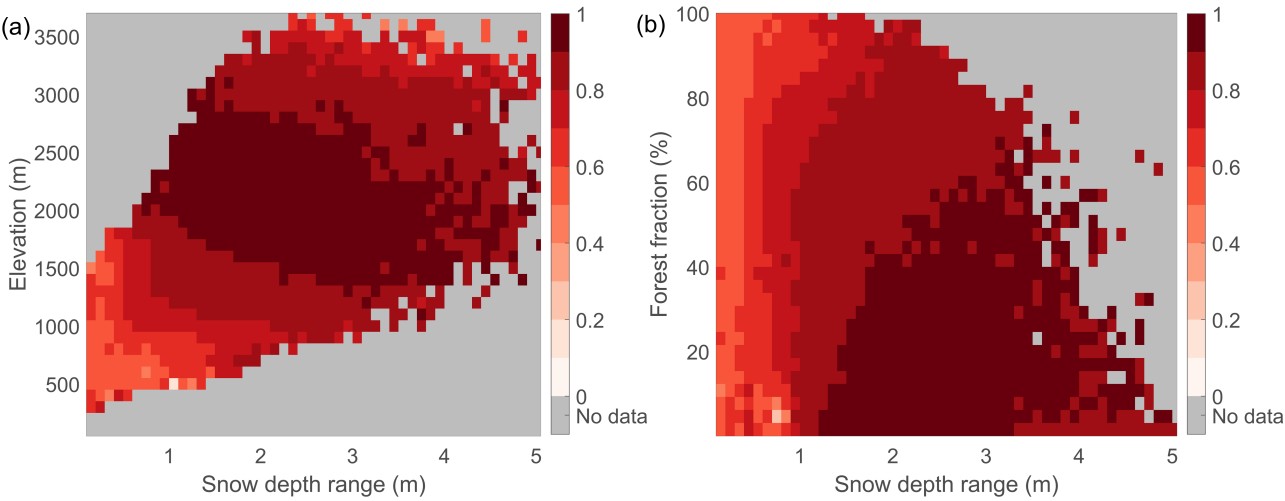

**Figure 6.** The time series correlation between weekly simulated and retrieved (dry) snow depth (m), stratified by (a) elevation (m) and snow depth range (m), and (b) forest cover fraction (%) and snow depth range (m).

results shown in Figs. 2–4, the correlations for areas with a maximum snow depth above ∼1 m are considerably higher for $\sigma^0_{VH}$ (Fig. 5b) than for $\sigma^0_{VV}$ (Fig. 5a), often reaching values above 0.6. The time series correlation is generally higher for S1 snow depth (Fig. 5c) than for $\sigma^0$, with values often reaching above 0.8. This higher correlation results from the use of a ratio between $\sigma^0$ in cross- and co-polarization as input to the change detection algorithm and from the re-initialization (i.e., HS=0)

at the start of every snow season, which removes differences in $\sigma^0$ levels between years. Fig. 5d further shows the correlation between the simulated and S1 snow depth with the inclusion of zero snow depths. This results in even higher correlations mainly in regions with shallow and occasional snow (e.g., in eastern Austria). Nevertheless, it is important to remark that the S1 $\sigma^0$ observations in these regions only show a weak (if any) correspondence with snow depth, because of the weak volume scattering contributions from shallow snow, frequent wet snow and melt conditions, and the frequent disappearance and re-

appearance of snow cover. The higher correlation values are therefore mainly attributed to the effectiveness of the IMS snow cover data set, used as auxiliary input in the retrieval, to discern between the absence and presence of snow, and should thus not be attributed to the S1 observations.

Figure 6 stratifies the time series correlation of Fig. 5d by snow depth range (based on the model simulations), as well as elevation (from SRTM) and forest cover fraction (from the Copernicus PROBA-V dataset). The correlation generally exceeds

0.7 when the range in snow depth exceeds ∼1 m, except in areas with an elevation below ∼1000 m or a forest cover fraction above ∼80%. The S1 retrievals are thus expected to be more accurate for the high snow regime (with snow depths reaching above 1 m), and in areas with lower forest cover fraction.

The S1 snow depth retrievals depend on a scaling factor $E$, which is either constant ($E_1$), or spatially varying ($E_2$), and based on model simulations. Figure 7 shows the scaling factor $E_2$ stratified per elevation band and per land cover class (see

Section 2.2). The optimization of the scaling factors by elevation indicates a correction towards lower snow depth retrievals

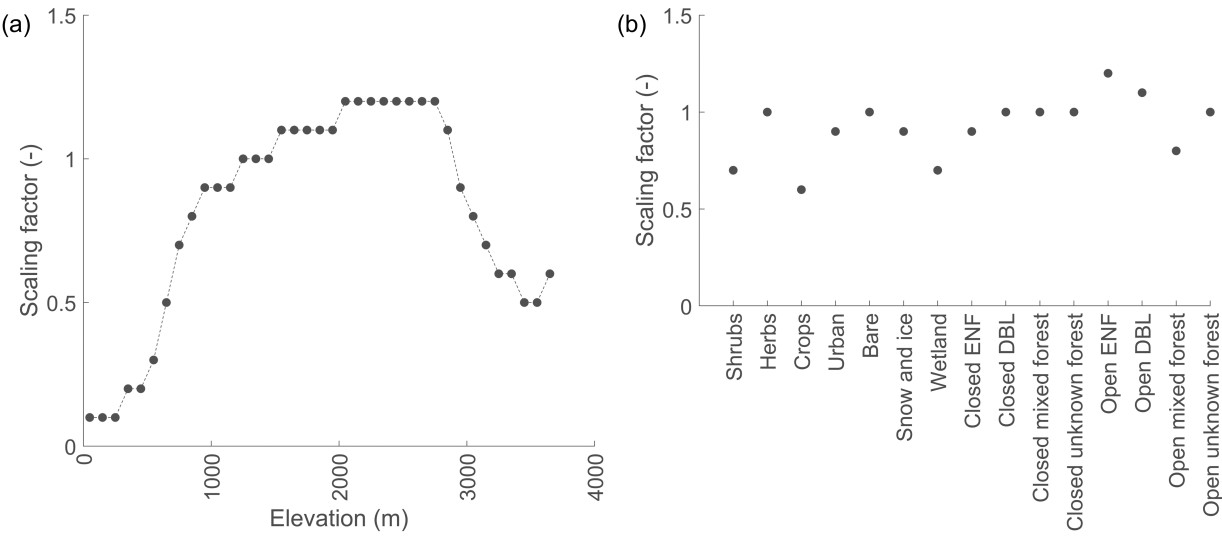

**Figure 7.** Scaling factor ($E_2$) for the conversion from SI (dB) into snow depth (m), stratified by (a) elevation and (b) land cover class. The scaling factors are derived after applying the constant scaling factor ($E_1 = 0.56\,\mathrm{m\,dB^{-1}}$; see Section 2.2).

at low elevations (below ∼1000 m), but also at high elevations (above ∼3000 m). At the low elevations, the signal-to-noise-ratio of the snow volume scattering is expected to be low due to the typically shallow snow conditions. Therefore, noise (or unaccounted impacts in the retrieval) can lead to an overestimation. At the high elevations, the lower scaling factors mainly address the overestimation in glaciated areas, where strong increases in the $\sigma^0$ observations may not only be caused by snow

accumulation, but also by the winter freezing of glacial meltwater. The optimization of the scaling factors by land cover indicates that generally slightly higher values are favorable in forested areas. This addresses the potential underestimation caused by the attenuation of the snow signal by the forest cover.

Figure 8 illustrates for every weekly time step from November through March the spatial correlation between the model simulations of snow depth and the corresponding S1 retrievals (obtained with the constant scaling factor $E_1$ and with the

10 spatially dynamic scaling factor $E_2$). It demonstrates the overall similarity in the spatial snow depth distribution for simulations and retrievals, with correlation values generally in the range of 0.6–0.9. The use of the spatially dynamic scaling factors results in minor improvements, that are largest with respect to the OSHD simulations in autumn 2018. Note that the dynamic scaling factors vary in space, but are constant in time, which means they do not affect the time series correlations (e.g., shown in Fig. 5 and 6). The spatially dynamic scaling factors are used in the retrievals of the next section on the validation with in situ

measurements.

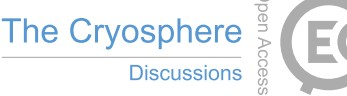

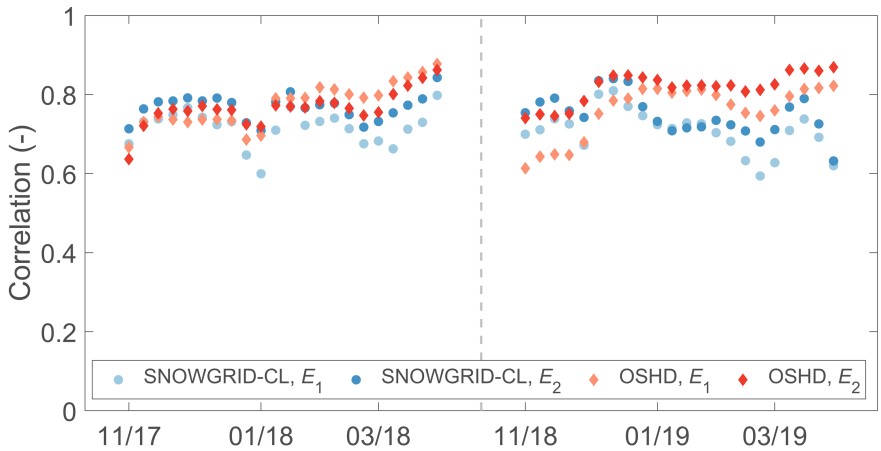

**Figure 8.** Spatial correlation between weekly S1 retrieved and simulated snow depths, using either spatially constant ($E_1$) or dynamic ($E_2$) scaling factors in the S1 retrievals.

### 3.3 Validation of sub-kilometer S1 retrieval with in situ measurements

Figure 9 shows examples of the S1 snow depth retrievals over the Alps at the 300 m spatial resolution for the weeks centered around 10/11/2017, 26/01/2018 and 01/03/2019. The S1 retrievals for instance capture the strong increase in snow depth during winter 2017–2018 (Fig. 9a–b). The Alps experienced several episodes of extreme snowfall in January 2018, caused by

a low-pressure system over the western Mediterranean that brought moist air northwards and resulted in an anomalously deep snowpack. The winter 2018–2019 was generally warmer than average, caused by a persistent high pressure system over western Europe. The warm temperatures led to extensive wet snow conditions during February, that were detected by S1 and used for masking the corresponding snow depth retrievals (Fig. 9c). Unlike in the northern Alps, where the wet snow occurrence is widespread on 01/03/2019, wet snow is mostly limited to south facing slopes in the southern Alps.

The S1 snow depth retrievals at the three different spatial resolutions (100 m, 300 m and 1 km) are compared against independent point-scale in situ measurements over the periods August through March 2017–2018 and 2018–2019 (Fig. 10). Although the more detailed retrievals at the finer 100 m resolution have the potential to better represent the local point-scale conditions at the in situ sites, a higher accuracy is obtained at the coarser 300 m and 1 km resolutions. The lower accuracy for the finer-scale retrievals can be caused by the larger impacts of: (i) radar speckle that is inherent to radar measurements, (ii)

geolocation errors and geometric distortions (foreshortening, layover) in the radar images causing location mismatches in $\sigma^0$ time series, and (iii) local heterogeneity in topography (elevation, slope, aspect), land surface properties (land cover, soil moisture, soil temperature and surface roughness) and snow variables (stratigraphy and microstructure). The 300 m retrievals offer a potential balance between resolution and accuracy. The resolution is sufficiently high to capture the high spatial variability in snow depth at the hill-slope scale, while the accuracy (spatio-temporal R=0.85, MAE=0.18 m) is only slightly lower com-

pared to the coarser 1 km scale (R=0.87, MAE=0.17 m). However, the resolution requirement may depend on the envisaged





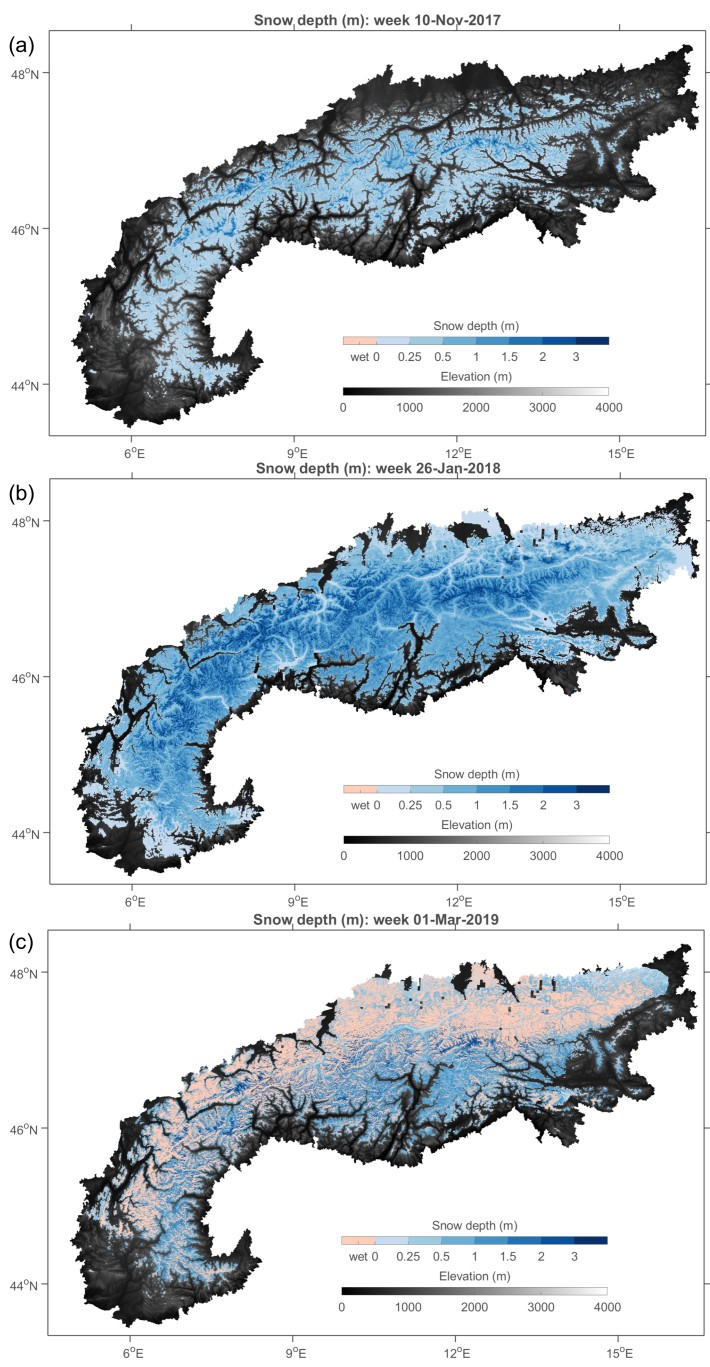

**Figure 9.** S1 snow depth (m) retrievals at 300 m spatial resolution over the Alps, for the weeks centered around (a) 10/11/2017, (b) 26/01/2018 and (c) 01/03/2019.





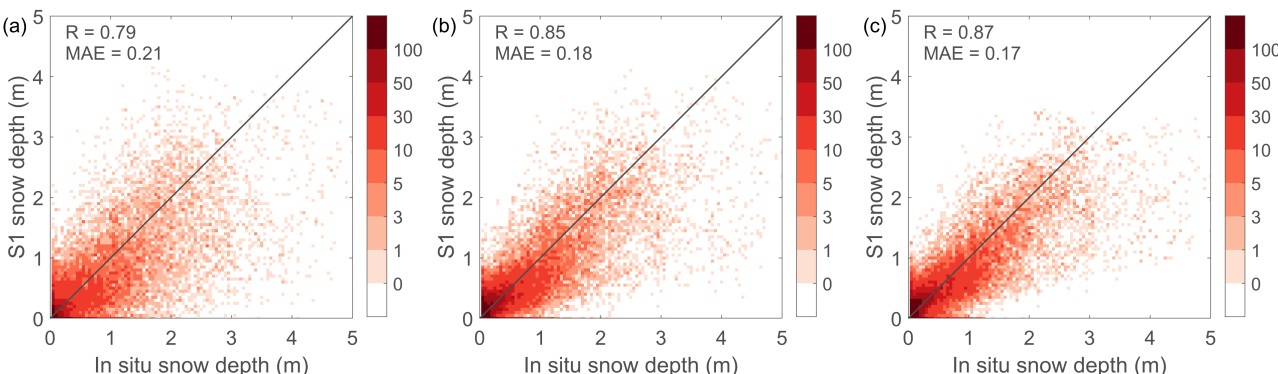

**Figure 10.** Density plots comparing weekly retrievals of snow depth (m) with corresponding in situ measurements (m) for August through March (wet snow excluded) 2017–2018 and 2018–2019. The comparison is shown for retrievals at (a) 100 m, (b) 300 m and (c) 1 km resolution. The spatio-temporal Pearson correlation R (-) and MAE (m) are shown on top left.

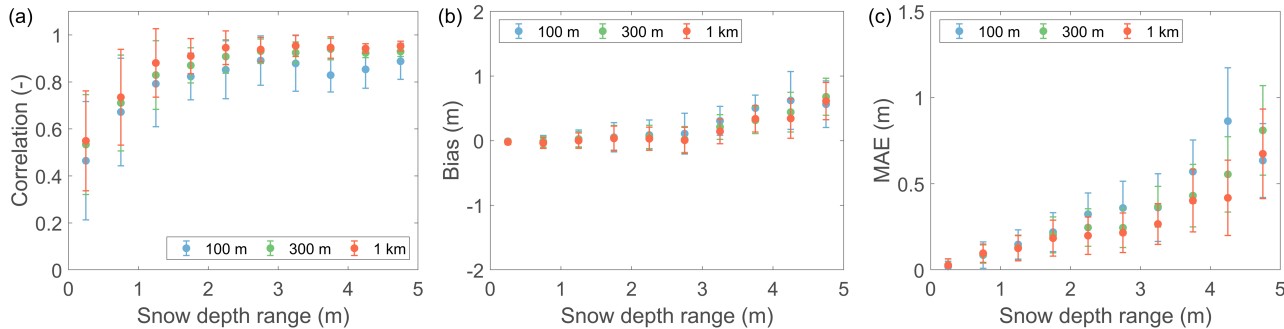

**Figure 11.** Spatially averaged time series validation metrics (± spatial standard deviation), comparing 100 m, 300 m and 1 km S1 retrievals with in situ measurements for August through March (wet snow excluded) 2017–2018 and 2018–2019: (a) Correlation (-), (b) Bias (m) and (c) MAE (m). The metrics are stratified by the snow depth range of the measurements.

application. For instance, the 1 km resolution may be too coarse for operational water management, but more than sufficient for regional numerical weather prediction.

The comparison between the S1 retrievals and in situ measurements is further stratified by the range in snow depth (Fig. 11). In agreement with the comparison against model simulations (Section 3.1), the time series correlation increases with the

5    snow depth range, reaching consistently high values (0.8–0.9) for sites with maximum snow depths above 1 m. The retrievals are mostly unbiased for sites with a maximum snow depth up to ∼3 m. A slight and increasing underestimation in the S1 retrievals occurs for sites with maximum snow depths ranging from 3 m to 5 m. Note that, at the same time, these higher snow depth measurements are less likely to represent the average conditions within the S1 grid scale. Similarly as the bias, the MAE increases with the snow depth range, and is on average estimated at ∼10–15% of the range. In general, the skill improves more

10    when retrievals are aggregated from 100 m to 300 m, than from 300 m to 1 km.





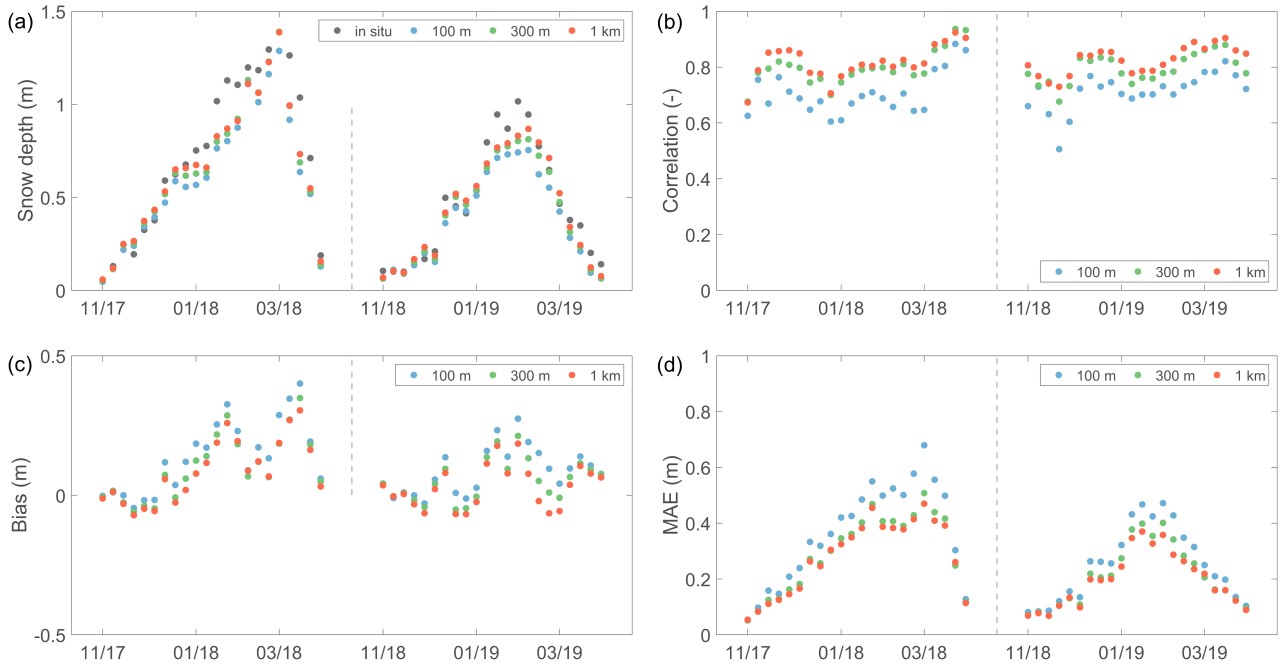

**Figure 12.** Spatial validation metrics, comparing 100 m, 300 m and 1 km S1 retrievals with in situ measurements for November through March (wet snow excluded) 2017–2018 and 2018–2019: (a) Mean snow depth (m), (b) Correlation (-), (c) Bias (m) and (d) MAE (m).

Figure 12 shows the evolution of the weekly spatial skill metrics over time, from November through March 2017–2018 and 2018–2019. Similar to the comparison against model simulations, the spatial correlation (Fig. 12b) between in situ measurements and retrievals in a given week slightly increases with the increase in average snow depth through winter, from values around ~0.6-0.7 in November to >0.8 in February–March. For most time steps, the bias (Fig. 12c) is slightly positive (i.e., in situ measurements are higher than the retrievals), which is caused by the scaling from SI to snow depth based on model simulations and not based on in situ measurements (Section 2.2). Moreover, in situ sites typically correspond with open flat field locations and often overestimate the mean snow depth of the surrounding terrain (Grünewald and Lehning, 2015). The temporal evolution of the MAE closely follows the evolution of the spatial average snow depth, with an increase in MAE with increasing snow depth. Overall, the S1 retrievals closely represent the evolution of the average snow depth (Fig. 12a) at the in situ sites during both winter seasons, as well as the inter-annual difference (more snow in the first winter).

## 4 Conclusions

Sentinel-1 (S1) backscatter observations, particularly in VH-polarization, correlate well with regional model simulations of snow depth over Austria and Switzerland. Using a combination of cross-polarized and co-polarized backscatter as input to a change detection algorithm, S1 snow depth retrievals at 100 m, 300 m and 1 km spatial and weekly temporal resolution





are presented over the European Alps for the period from August 2017 through July 2019. The accuracy of the retrievals is validated using independent local time series measurements of snow depth at 743 locations across the Alps. Despite the typically large representativeness differences between these point-scale in situ measurements and grid-scale satellite retrievals, good skill metrics are obtained. At the 300 m resolution and for dry snow conditions within the months August through
March, the spatio-temporal Pearson correlation coefficient is 0.85 and the mean absolute error 0.18 m. A slightly higher (lower) accuracy is observed at the 1 km (100 m) resolution. The lower accuracy at the fine scale is likely caused by measurement noise, geolocation errors and geometric distortions, and the local heterogeneity in topography, land surface and snow properties.

The main uncertainties in the S1 snow depth retrievals are expected to be caused by wet, shallow and occasional snow cover and forest cover. Wet snow is known to cause a strong decrease in radar backscatter due to signal absorption. Although a wet
snow detection algorithm is implemented, undetected wet snow (for instance due to an insufficient decrease in the backscatter) may cause underestimation in the snow depth retrievals. Uncertainties can also be large in regions with shallow and occasional snow cover, where the backscatter observations can be dominated by scattering contributions from the ground surface, resulting in a weak (or even negative) correlation with snow depth. For shallow snow conditions, backscatter observations at higher frequencies (e.g., X- or Ku-band), or potentially also using InSAR phase changes at lower (e.g., L- or P-band) frequencies,
could be more suitable to detect short-term snow depth changes. Higher uncertainties in the S1 retrievals may also be found in regions with large open water (lake) areal coverage, and in densely forested regions, where the attenuation of the radar signal may reduce the sensitivity to snow. The comparison of S1 retrievals with regional model simulations of snow depth reveals that sensitivity is generally strong (e.g., time series correlations >0.7) if the maximum snow depth reaches above ∼1 m, at an elevation above ∼1000 m or with a forest cover fraction below ∼80%.

The Sentinel-1 retrievals can provide important information on the spatial distribution of snow depth in regions with complex topography, which are typically excluded in passive microwave retrievals. Therefore, the results of this study may contribute to improving our knowledge on the terrestrial snow mass budget. An asset of the approach is the confirmed long-term availability of S1 observations, with continuity by S1C and S1D over the coming decades. This will allow for generating the long time series that are required for investigating the potential impacts of climate variability or climate change on snow mass in mountain
regions and for assessing the corresponding impacts on water availability to society. Finally, the combination of the S1 retrievals with land surface model estimates through data assimilation is expected to be rewarding. Not only could the assimilation ensure improved and continuous (in time and space) estimates of mountain snow mass, it is also likely to benefit associated applications such as operational water management, natural hazard (avalanche and flood) prediction or numerical weather prediction.

*Data availability.*

The Sentinel-1 snow depth retrievals at 300 m and 1 km spatial and weekly temporal resolution are available online at https://ees.kuleuven.be/project/c-snow.



*Author contributions.*  H. Lievens, I. Brangers, H.-P. Marshall and G. De Lannoy contributed to the Sentinel-1 snow depth retrieval algorithm. H. Lievens performed the analysis. T. Jonas and M. Olefs produced the snow model simulations and provided in situ measurements. All co-authors contributed to the writing of the manuscript.

*Competing interests.*  No competing interests are present.

5  *Acknowledgements.*  This work was funded through the BELSPO C-SNOW project. Sentinel-1A/B data are from the ESA and Copernicus Sentinel Satellites project and were processed using the ESA Sentinel Application Platform (SNAP). The resources and services used in this work were provided by the VSC (Flemish Supercomputer Center), funded by the Research Foundation - Flanders (FWO) and the Flemish Government.



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
