# Peer review of "Sentinel-1 snow depth retrieval at sub-kilometer resolution over the European Alps"

_The Cryosphere, 2021_

## Author Response (AR1)

Title: Sentinel-1 snow depth retrieval at sub-kilometer resolution over the European Alps
Authors: Hans Lievens, Isis Brangers, Hans-Peter Marshall, Tobias Jonas, Marc Olefs, Gabriëlle De Lannoy
Submitted to: The Cryosphere

*Author responses below are in italic. Page and line numbers refer to the revised manuscript version with changes marked in bold font.*

**1. Response to editor comments**

Comments to the Author:

Dear Dr. Lievens:

Thank-you for your responses to the reviews of your manuscript.

There is clearly a difference in perspectives with Reviewer 1, which need to be addressed in the revised manuscript. Reviewer 1 accepts the potential for C-band backscatter sensitivity to the dry snow volume of deep mountain snowpacks (through the probability for greater snow anisotropy and hence greater scattering and depolarization of the SAR signal in deeper snow) but recommended rejection of the manuscript based on the level of processing of the Sentinel-1 imagery ("With all the processing done to the SAR imagery, it is impossible to assess the physical interactions of the SAR signal with the snowpack since the data has been smoothed multiple times and transformed radiometrically and geometrically.") Reviewer 1 also identified a series of issues with the statistical approach used in the validation (including the treatment of outliers and zero values) which will require new calculations. I was pleased to see your response to the important comment regarding open data standards.

Reviewer 2 recommended major revisions, essentially asking for a reworking of the manuscript to clearly communicate the lack of understanding of physical mechanisms, the empirical nature of the retrieval, and the impact of calibration.

Both reviewers positively note the amount of work that has been done to mine the Sentinel-1 dataset, the overall quality of the writing, and the potential impact of this work across the snow and hydrology communities. It is the very nature of this high impact which means the Sentinel-1 data processing decisions, a clear description of the physical mechanisms (or lack thereof), robust validation, and a clear description of the empirical nature of the retrieval (including the role of calibration) should be made more explicit in the manuscript. Your response to the reviews indicates agreement on most of these points.

A public comment (to which you have already submitted a reply) also expressed reservations similar to those raised by the two reviewers. It is my interpretation that revision of the manuscript to meet the issues raised by the reviewers will also go a long way to addressing the concerns of this comment.

My assessment is that major revisions are required and the manuscript will be returned for an additional round of review, after which a final editorial decision will be taken.

Thanks very much for your contribution to The Cryosphere Discussions.

Chris Derksen

*Dear editor, we'd like to thank you for the assessment of our manuscript. We are also thankful to the reviewers for providing a detailed assessment. We have modified the manuscript substantially to account for the reviewer comments as summarized below, which we believe has improved the quality of the work. The public comments by Helmut Rott were much appreciated by the authors and have also been accounted for in the associated response and/or in the revised manuscript.*

*Based on the recommendations of reviewer 1:*

- *We fully re-processed the S1 backscatter data, in order to perform the terrain flattening and terrain correction at high resolution (similar to that of the DEM), and to output backscatter as gamma0.*
- *The number of backscatter post-processing steps has been reduced. More specifically, the rescaling of sigma0 based on the incidence angle was no longer required by the use of gamma0, and the outlier correction has been deactivated.*
- *Using the newly processed gamma0 data, all analyses steps (e.g., snow depth retrieval, validation) have been repeated.*
- *The theoretical discussion has been extended to include the potential impacts of snow anisotropy and multiple scattering on layer interfaces, and future research has been recommended to investigate the impact of snow microstructure and stratigraphy (also in response to the comments by H. Rott)*
- *We have revised the validation to clearly distinguish between spatial and temporal correlation (Fig. 11; also in response to the comment by H. Rott), and to provide validation metrics for both scenarios with and without zero snow depths included. Further, the mean absolute error and bias are stratified by the measured snow depth in Fig. 12., and presented both as absolute and relative values.*
- *The wet snow detection algorithm was extended and modified to start earlier and to allow the calculation of fractional wet snow cover. This allows the user to define the preferred level of wet snow masking.*

*Based on the recommendations of reviewer 2:*

- *We now explicitly mention the empirical nature of the retrieval algorithm in the abstract, introduction and conclusions.*
- *Similarly, the lack of physical understanding about why C-band SAR data are sensitive to snow depth is highlighted at various places in the manuscript, including the abstract, introduction and conclusions.*
- *We have discussed more in detail the scaling of the retrievals (calibration) based on reference data and its implications to apply the method in other regions.*
- *We highlighted the need to further test and validate the method in other regions in a dedicated paragraph at the end of the conclusions.*

*As a result of the above-mentioned modifications, the performance of the retrievals and validation analysis has improved. The wet snow masking component is more evolved and allows a more flexible choice in terms of the preferred level of wet snow masking. The discussion on the physical basis of backscatter sensitivity to snow has been extended, also clearly identifying the need for further research. To allow anonymous access to the data during the manuscript review, we here provide the login details to the server:*

*Server:      sftp://hydras.ugent.be*
*Login:       sentinel1snow*
*Password:    s1!csnow*
*Port:        2225*

*We are hopeful that these modifications adequately address all the comments by the reviewers. Please find a point-by-point reply to each of the comments below.*

*Sincerely, Hans Lievens, on behalf of all co-authors.*

**2. Response to Reviewer 1 comments**

This paper builds on the work of Lievens et al., 2019 to extract snow depth from S-1 data in the Alps. As mentioned by the editor, this work is of high relevance to the snow community but also to many other research areas such as water management, tourism, climate change and biodiversity. I appreciate the work that is done here but in its current state, I cannot recommend this paper for publication since I feel there are too many unknowns and too much processing done on the S-1 imagery to be able to retrieve some sort of good quality snow information and give a proper assessment of the results shown here. This is reflected in my comments below.

Contrary to what has been stated by the authors in their response to the editor's comments, I am not skeptical of the relationship between the C-band signal and thick alpine snowpacks. I do question the physics of the approach used in this study and am concerned about the multiple layer of data smoothing in order to get good correlations with modelled data.

If the authors are willing to provide more information on the imagery processing and modify it to make it more physically accurate, I strongly believe this work has great value to the scientific community.

*We would like to thank the reviewer for the detailed assessment of our work. To address the reviewer's main concern about the processing of the S-1, we have carried out a full re-processing of the S-1 data across the Alps with a revised methodology (see details below), without any significant change in the results or conclusions.*

*We however strongly disagree with the statement that the good correlations with modelled data are due to the multiple layers of data smoothing. The smoothing applied here is limited, as discussed below.*

*We are hopeful that the revised processing of the S-1 data and a more detailed discussion of the processing and algorithm steps (including some modifications, e.g. regarding the wet snow detection) adequately address the main concerns of the reviewer.*

General Comments:

As mentioned above, I do agree with the authors that the cross-pol channel of S-1 can be sensitive to a thick snowpack but I disagree with the physical explanation of the authors. The physical interaction of the microwave signal with the snowpack is very complex and is not solely related to surface/volume scattering and single/double bounce. With snow layer thicknesses close or smaller than the wavelength, you have many interference and coherence effects in the signal. Recent work has shown that volume scattering and depolarization of the SAR signal comes mostly for the snow anisotropy (Leins et al., 2016) and the vertical/horizontal structuring of the snowpack at C-band. This can be achieved by a stratified snowpack horizontally or with snow grains that are structure vertically/horizontally through metamorphic processes. I would agree that with a thicker snowpack, chances are you will get more anisotropy but this is not shown with in situ measurements, temporal analysis or snowpack stratigraphic information.

*We appreciate the reviewer's comments on the physics of the signal. Although the mentioned work by Leinss et al. (2016) investigates only higher (X- and Ku-band) frequencies, we agree that the anisotropy of snow crystals and/or of crystal clusters, as well as the snow stratigraphy, can play an important role.*

*The manuscript has been modified as follows:*

*P3, L2: "More specifically, dry snow represents a multi-layered, dense medium of irregularly shaped (anisotropic) ice crystals that can form larger-scale clusters. Signal depolarization can for instance occur due to scattering within the dense anisotropic snow volume, multiple scattering on snow layer interfaces and snow-ground scattering interactions (Du et al., 2010; Chang et al., 2014; Leinss et al., 2016)."*

*P3., L29: "Future research is recommended to further investigate the physical scattering mechanisms in snow at C-band, including the impacts of snow microstructure and stratigraphy, …"*

*P5, L21: "The snow depth retrieval algorithm is based on the assumption that a snowpack is typically composed of multiple snow layers, where each of the layers represents a dense medium of clustered, irregularly-shaped (anisotropic) ice crystals within an air background. We hypothesize that the microwave signal in the snowpack depolarizes primarily by the scattering on anisotropic clusters of snow crystals within the snow volume (Chang et al., 2014), by the multiple scattering between snow layer interfaces (Du et al., 2010), and by snow-ground scattering interactions. A deeper snowpack is likely to result in more opportunities for signal depolarization and therefore stronger scattering in cross-polarization."*

*P13, L5: "The dry snow scattering contribution is stronger, most likely because depolarization occurs after volume scattering on anisotropic crystals or clusters of crystals, multiple scattering on snow layer interfaces, and snow-ground scattering interactions."*

With all the processing done to the SAR imagery, it is impossible to assess the physical interactions of the SAR signal with the snowpack since the data has been smoothed multiple times and transformed radiometrically and geometrically. You have multi-looking (averaging 10x10 pixels), border noise removal, thermal noise removal, terrain correction and reprojection to the WGS84 projection. The multi-looking is especially concerning given the topographic complexity of the Alps. It is smoothing all the topographic information (which is crucial for snow retrievals) and emphasizing only the areas of significant snow (snow drifts) which is not representative of a 100m grid cell in the Alps. Then you add incidence angle correction using a DEM (30m) that is of lower resolution than the pixel spacing (10m) of the original image. A DEM with similar resolution should be used but also, the topographic information has already been altered from the multi-looking which is not representative of the local topography. Then there's temporal averaging (Eq.2) which alters the signal even further. Finally, outliers are replaced by a 12-day average to smooth the data once more.

*We would argue for the contrary: a careful processing is a pre-requisite in order to assess the correspondence between the S-1 signal and snow depth. The processing steps included in our analysis (border noise removal, thermal noise removal, multi-looking, terrain correction and reprojection to a consistent grid) are all standard and necessary procedures, recommended by any manual or handbook on SAR processing. The multi-looking is arguably the only processing step that could be considered optional. However, this was included in order to (i) reduce the impact of radar speckle, (ii) reduce the processing time (note that more than 4000 S-1 images were processed), and (iii) reduce the data storage requirements. In this context, the multi-looking is an important step to keep the processing computationally feasible also for larger areas, not limited to the Alps.*

*However, to address the reviewer's comment about the correction with the DEM, we have re-processed the S-1 data over the Alps, by performing the range-Doppler terrain correction and terrain flattening at the 20 m S-1 resolution instead of at the multi-looked 100 m pixel spacing. We kept the original 30 m DEM (SRTM 1Sec HGT) because this is the standard suggested DEM for processing in the ESA SNAP toolbox and can also be applied in other regions (e.g., outside Europe, or where more detailed DEM information is lacking). However, the pixel sizes of the DEM and the S-1 data were now much more similar with the re-processing. Equation 2 (Equation 1 in the revised manuscript) is not performing temporal averaging as stated by the reviewer. It applies a bias correction (of the first two order moments, i.e., the mean and variance) to every individual backscatter observation, without averaging observations over time. The bias correction reduces the differences between observations from different orbits (e.g., caused by different incidence or azimuth angles) and we strongly recommend this step for any application that aims at combining information from different S-1 orbits. We have deactivated the outlier correction in the retrieval because we observed it was slightly interfering with the wet snow detection algorithm.*

Further on the processing, I would avoid talking about sigma-nought when Eq. 1 converts the sigma-nought into a pseudo-gamma-nought multiplied by cos(40). I say pseudo here because the incidence angle used to

convert sigma-nought is the 100m reprojected angle and not the gamma-nought values from the SAR imagery calibration.

*At the time of the S-1 processing for our initial manuscript submission, the calculation of gamma0 was not operational in the SNAP software version 7. In the revised processing, we appropriately calculated gamma0 using SNAP version 8, by first calibrating the backscatter observations to beta0 and subsequently applying terrain flattening. The entire analysis, including the snow depth retrieval and validation, has been repeated with the use of backscatter as gamma0 and is thus more conform the state-of-the-art.*

If we accept the processing chain of the SAR imagery, it is still unclear that what the correlations are showing is linked to the snow depth. The errors obtained from the SAR retrievals (Figure 11) are most of the time larger than the precision of the reference data which is the model simulations. It is very difficult to determine that the correlations are statistically significant in this case and also looking at Figure 10, most of the comparison points are grouped around 0 which tends to falsely boost the correlation.

*We do not see any reason to not accept the processing chain, especially considering the re-processing discussed above, which is fully compliant with the state-of-the-art. Figure 11 shows a comparison against in situ measurements as reference data (not model simulations). We are surprised that the reviewer questions the significance of the time series correlations in Figure 11, which are typically above 0.8 (for sites that feature snow depths thicker than a meter).*

*With respect to Figure 10, we agree that the abundance of low snow depths impacts the correlations. To identify this impact, we have calculated and reported validation metrics (R, MAE) for scenarios both with and without inclusion of zero snow depth values in Figure 10. Even though many data are clustered around low snow depths, the density plots in our opinion still clearly demonstrate the overall agreement between the S-1 retrievals and the in situ measurements also for the high snow depths, especially for the coarser 300 m and 1 km retrievals. The revised Figure 12 supports this statement, showing that the lowest relative errors are obtained for snow depths of about 2-2.5 m.*

Given that modelled data is often smoothed and often have difficulty capturing extreme snow conditions and that the SAR data has been smoothed many times and outliers replaced by temporal means, I can't say I am surprised to see a good empirical relationship.

*We respectfully disagree with this comment, as the S-1 processing does not include multiple smoothing steps, unlike what the reviewer states (see above). In our opinion, the strong relationship between the S-1 retrievals and the model simulations is encouraging, and is furthermore corroborated by the strong correspondence between the S-1 retrievals and in situ measurements.*

Also, asking scientists to identify themselves in order to get access to the data used in this study does not comply with the open data policy.

*We understand this comment. To share the snow data over the Alps, we have used the existing platform via which we also share the corresponding retrievals across the Northern Hemisphere mountains. Please note the following login details, which allow to directly access the ftp server anonymously:*

*Server:        sftp://hydras.ugent.be*
*Login:         sentinel1snow*
*Password:   s1!csnow*
*Port:           2225*

Specific comments:

P.3L.5: I would disagree with the claim that an increase snow depth automatically causes an increase in volume scattering. If their is not sufficient anisotropy in the snowpack, there will not be any volume scattering in C-band. The theory will show that even if you increase the snow depth and keep all other snowpack parameters constant, you will not have a significant increase in volume scattering

*We have improved the formulation of the potential mechanisms that can enhance the scattering in cross-polarization from the snowpack (see above). Please also note that recent radiative transfer model simulations using Bic-DMRT have shown that cross-polarized backscatter at C-band can increase with an increase in SWE (or depth) while keeping other parameters (snow grain size, snow clustering) constant (personal communication with Prof. L. Tsang, University of Michigan).*

P.3L.6: Again, this comment is highly dependent on the stratigraphy and anisotropy of the snowpack. This section needs to be supported by snowpit measurements of the studied area or referred to past work done in the area analyzing the snowpack properties.

*The statements on P.3L.6 were general assumptions based on which the empirical change detection retrieval approach is built. We have not yet analyzed these assumptions using snowpit measurements as suggested by the reviewer, but this is foreseen in future research. However, Figures 3 and 4 (based on model simulations) support the statements that (i) an increase in snow depth generally increases (especially cross-pol) backscatter, that (ii) the snow scattering (in cross-pol) is not negligible compared to the ground scattering, and (iii) that ground surface properties remain relatively constant in time due to the insulating properties of snow, thus the main changes in backscatter over time relate to changes in the snowpack.*

*The need to further investigate the impacts of snow microstructure (including anisotropy) and stratigraphy has now been highlighted at several places in the revised manuscript (e.g., P3, L31; P23, L17).*

P.3L.7: This comment is most likely true for the studied area but again, no reference or field measurement is provided to support this claim.

*Please refer to the response above.*

P.3L.9: Again here, I strongly disagree with this claim. The microstructure, anisotropy changes and stratigraphy, especially in the bottom layers of the snowpack will most likely drive the changes in sigma0.

*We aim to further investigate the impact of microstructure and stratigraphy in future research, based on tower-mounted radar measurements currently being collected in the Rocky Mountains, US. For now, we have generalized the statement in the revised manuscript to "the main changes of backscatter over time can be related to changes of the snowpack".*

P.3L.30: Even though this is common processing of SAR imagery, this is considerably altering the SAR signal, considerably smoothing it and making it very difficult to link to any ground snow properties.

*The alternative (i.e., not performing thermal noise removal, border noise removal, radiometric calibration, and terrain correction) would lead to inferior processing results, which we believe would be far less suitable to investigate the relationship between backscatter and snow depth. The multi-looking has been adjusted and is not impacting the terrain correction and terrain flattening in the re-processing. Please also refer to our responses above.*

P.3L.32: Multi-looking (or block averaging here) is a good way to reduce speckle noise in flat terrain. Here though, the topography is very complex (as mentioned by the authors) and it is emphasizing on the geometric

distortions and the areas of significant snow (snow drifts) which is often not representative of a 100m grid cell in alpine areas.

*Please refer to our responses above.*

P.4L.10: Using "local" incidence angle correction on a multi-looked image is not an accurate method. A DEM with similar resolution as the raw image should be used to correct for local incidence angle before multi-looking.

*This has been addressed by the reprocessing to gamma0 with terrain correction and flattening being applied at the 20 m S-1 resolution.*

P.4L.15: This relationship was developed for areas of flat terrain and is not representative of the studied area. Proper analysis of the backscattered signal as a function of local incidence angle needs to be conducted in alpine areas in order to find the proper normalization relationship. A before and after image should show that this is not normalizing the image properly. Also, this is exactly taking σ-nought and converting it to gamma-nought and then multiplying it by cos(40).

*This comment has been accounted for by processing to gamma0.*

P.4Eq.2: Here again, temporal smoothing of the data. There's no way of linking the spatio-temporal snow properties of the original SAR imagery.

*We disagree. Equation 2 (now Equation 1) is not performing temporal smoothing, but bias correction, which results in an improved S-1 processing quality and therefore benefits the analysis with respect to snow depth.*

P.4L.27: Excluding March to July is very subjective here. First, it is removing a lot of snow properties variability which can occur in March. Anisotropy and stratigraphy is stronger in the later winter season. Second, with climate change, we know that wet snow is detected outside of this period.

*We are convinced that the remaining period, i.e., from August through February for two consecutive years, includes sufficient backscatter observations to allow an accurate calculation of the temporal mean and standard deviation, which are needed for the orbit bias correction. As mentioned in the manuscript, the reason why we exclude March to July is to avoid strong impacts from wet snow on the calculation of the mean and variability, and thus on the backscatter processing for dry snow conditions that are most important for the retrieval. We fully agree that wet snow can also impact the observations earlier than March. Therefore, we have revised the wet snow detection in the retrieval algorithm, to not limit the detection only to the period from February onwards (see response below).*

P.4 L.30: This is not rigorous. Removing outliers is another method to smooth out the data and get better correlation with modelled data. But here they are not only removed, they are replaced by a smoothed average.

*We have deactivated the outlier removal, because we observed it was slightly interfering with the wet snow detection. More specifically, the outlier removal caused some wet snow events to be undetected, because the backscatter had been modified by the outlier correction.*

P.5 Eq.5: Is A applied to the ratio or only the cross-pol channel?

*A is applied only to the cross-pol channel, enhancing the sensitivity to snow depth which is primarily driven by the cross-pol observations.*

P.6L.17: I appreciate this approach where the index varies in time but I feel like the threshold is still limiting. I would see a temporal analysis of the SAR signal through multiple years to try and identify the proper threshold.

*We have tested a range of threshold values. The lower the threshold, the better wet snow impacts are reduced, however, at the expense of reducing the coverage. We identified a threshold of 1.5 to strike a balance between wet snow filtering and data coverage. Please refer to Figure 4 showing the effectiveness of the wet snow detection algorithm for two years at two locations (in Austria and Switzerland).*

P.6L.25: Again, the February start is very subjective as wet snow conditions can be detected earlier and the September-November period is most likely to be the period where you have the highest backscatter and all the values that are 3dB below might be because of small surface moisture or percolating water which is not uncommon in Alpine snow.

*In the revised version, we have activated the previous wet snow detection mechanism earlier (in January), and included an additional wet snow detection mechanism. The latter consists of (i) excluding backscatter observations (between the start of the snow season and end of December) that are a threshold (e.g., 2 dB) below the 10-percentile of backscatter observations during snow-free conditions, and (ii) excluding negative snow index values from January onwards. More specifically approach (i) improves the detection of early wet snow, often in autumn, whereas (ii) mainly improves the wet snow detection in the valleys, where a sharp decrease in backscatter during snowmelt is often lacking (e.g., due to forest cover). Furthermore, the wet snow is now provided along with the unmasked snow depth retrievals, allowing the user to choose whether or not to mask out wet snow, or to use another mask (e.g., derived from modeling or an alternative wet snow detection approach). In the 300-m and 1-km datasets, the wet snow is provided as a fraction (0-1) of wet snow pixels, which allows the user to define the level of wet snow allowed. Please refer to the manuscript (P7, L19) for details on the wet snow detection method. Results with various levels of wet snow masking are shown in the revised Figure 10.*

*Conversely to what the reviewer hypothesizes, the September-November period is typically the period in time with the lowest S-1 backscatter values, especially in cross-pol (if not including wet snow conditions in spring). Earlier (in summer), vegetation often contributes to higher backscatter, whereas in mid-winter, a higher backscatter is caused by snow accumulation. Part of the lower backscatter values in September-November can also be explained by the potential freezing of the soil surface, and/or by early wet snow.*

P.11L.7: There is no mention of layering and anisotropy which is most likely the main reason of signal backscattering of dry snowpacks.

*This has been included in the revised manuscript. Please refer to our response above.*

P.11L.11-13: These comparisons do not really apply to the current studies. As was mentioned by the authors in the response to the editor these studies were conducted in shallow snow conditions in tundra/taiga landscapes.

*The Alps include areas with shallow snow for which the references to literature are relevant. The literature comparison also helps to indicate that (to our best knowledge) studies with cross-pol observations in deep snow are lacking.*

P.11L.20: This is a strong assumption since in alpine regions you can have strong surface roughness that will depolarize your signal.

*The ratio of cross- over co-polarized backscatter is considerably lower in areas with limited vegetation. Hence, this statement is supported by S-1 observations.*

P.11L.33: This is normal since most of the volume scattering and depolarization will come from the forest cover. For this study, I would have masked out the forested areas because this adds unnecessary complexity to a study that is already complex. Masking the forested areas would allow to focus on the snow retrieval without getting confused in multiple empirical relationships and heavy data processing.

*One could either mask out the forested regions, or stratify the performance based on forest cover. We here opted for the stratified performance assessment (see Figure 6), which is more complete. We do not consent with the assessment of 'heavy' data processing.*

**3. Response to Reviewer 2 comments**

The authors present an application of a change-detection algorithm to estimate SWE in the Alps using Sentinel-1 C-band SAR. They explore the effect of spatial resolution on their retrievals. This is an important and timely contribution, and should be of great interest to the community. The paper is well-written so I have very few minor comments. Instead, I'll focus on a really key point which is that I think there is a great chance for readers to misunderstand the maturity level of the algorithm, based on how the paper is presented. This review is five related major comments that unpack this idea.

*We are grateful for the assessment of our work by the reviewer. Please find below our response to the five posted comments.*

Major Comments

First, I do not think that the paper adequately reflects the fact that we still do not understand why this method works, even at a basic level. The manuscript instead makes it sound clear that the mechanisms are understood: e.g. in the introduction, page 2, lines 32-page 3, line 2. Taking their points one by one: to their first point (page 2 line 33), no reference was given, and no reason why having lower ground backscatter would change sensitivity to depth; to their second point (page 2, line 33), Chang et al. 2014 do not make this point, that I could see. Readers will assume after reading the introduction that it is obvious why the C-band cross-pol is correlated with snow depth, which is not true. In fact, the authors of this study only introduce the idea that the "physical mechanisms that cause this increase are still uncertain" in the Results & Discussion section (page 10, line 13). Please, bring this critical point into the abstract, introduction and conclusion!

*We fully agree that we need to better inform the reader about the current limitations in physical understanding of C-band sensitivity to snow, upfront in the paper. We have highlighted this in the following sections:*

*Abstract (P1, L14): "However, future research is recommended to further investigate the physical basis of the sensitivity of Sentinel-1 backscatter observations to snow accumulation."*

*Introduction (P2, L33): "Although further research is needed to improve our basic understanding of the physical scattering mechanisms, we hypothesize that …"*

*Introduction (P3, L29): "Future research is recommended to further investigate the physical scattering mechanisms in snow at C-band, including the impacts of snow microstructure and stratigraphy, and to extend the validation over regions with different soil, vegetation and snow conditions, also using validation data at the matching scale of the satellite retrievals."*

*Conclusion (P23, L15): "Further research is needed to investigate more in depth the physical basis of C-band radar backscatter sensitivity to snow, for instance based on tower radar measurements and corresponding detailed measurements of snowpack properties, including snow microstructure and stratigraphy."*

*Regarding the statements on page 2, lines 32 to page 3, line 2: The first statement on page 2 line 33 ("surface scattering from the ground is significantly weaker in cross-polarization") refers to the common understanding that cross-polarized backscatter is typically several dB lower than co-polarized backscatter. This is especially the case in regions with limited vegetation and for smoother surfaces (vegetation and surface roughness increase depolarization and thus the cross-polarized backscatter, which, however, will generally still remain lower than the co-polarized backscatter). It is also common understanding that in a logarithmic (dB) scale as used in the retrieval algorithm, an increase in scatter intensity (in linear scale) will have a relatively larger impact when the prior intensity is low; hence the statement that a lower ground backscatter can be beneficial for the sensitivity to snow.*

*Regarding the reference to Chang et al. (2014): They mention the following statements in their introduction that support our quote (i.e., "dry snow represents a dense medium of irregularly-shaped and clustered ice crystals that primarily causes volume scattering in cross-polarization"): "In snow, the ice particles are packed closely together", "ice grains in snow do not scatter independently", "shapes are irregular and there are clustering effects", "In conventional scattering models, there is no cross-polarization in scattering when particles are spheres. In the dense media model, the electric dipole interactions of closely packed ice grains result in strong cross-polarization in the phase matrices".*

Second, I think it is critical to communicate more clearly throughout that this is an empirical algorithm with calibration parameters that require known SWE data over the domain. The word "empirical" needs to appear in the abstract, in my opinion. Please somehow get this idea into the introduction, abstract, and conclusion.

*We agree and have modified the text accordingly. The empirical nature of the retrieval algorithm has been mentioned at various places (P1, L7; P3, L10; P6, L3; P7, L3; P8, L14; P21, L15).*

Third, the authors need to point out that the algorithm only works well if you have accurate SWE data to calibrate against. Indeed, they need to just note explicitly that the accuracy of the approach they are using here is limited to the accuracy of their training data. I think this needs to be presented explicitly in the abstract and conclusions, to avoid reader misunderstanding.

*The approach indeed requires reference snow depth data in order to estimate the scaling coefficient that translates the changes in backscatter into changes in snow depth. However, we would like to highlight that the approach works already reasonably well when using a single, constant (both in time and space) scaling coefficient. For instance, Lievens et al. (2019) apply a constant scaling factor across all mountain ranges in the Northern Hemisphere, which still leads to relatively accurate retrievals. Therefore, the need for accurate reference data is not considered to be critical.*

*Figure 8 shows a positive but overall limited impact of refining the scaling coefficient, by allowing it to vary in space (but still not in time). Such refinement of the scaling coefficient may indeed require more accurate reference snow depth data, but again, this impact is limited.*

*The following text is incorporated in the manuscript to address this comment:*

*P7, L7: "Note that the scaling based on model simulations can propagate simulation uncertainties into the retrieval."*

*P7, L8: "As a first approximation, E1 is considered constant in space and time, conform with Lievens et al. (2019). This approach is applicable to mountain ranges for which no detailed information on the spatial snow*

*depth distribution, e.g., provided by model simulations, LiDAR data or dense networks of in situ measurements, is available."*

*P7, L16: "Note that such approach may improve the spatial distribution of the snow depth retrievals, but is only feasible in regions where sufficient reference information is available."*

*P23, L18: "Also, future research is recommended to investigate the validation of the S1 snow depth retrievals using data at the matching scale, for instance from LiDAR, and to further investigate the performance of the approach in other regions with different soil substrate types, vegetation conditions and snow climate conditions."*

*P3, L29: "Future research is recommended to further investigate the physical scattering mechanisms in snow at C-band, including the impacts of snow microstructure and stratigraphy, and to extend the validation over regions with different soil, vegetation and snow conditions, also using validation data at the matching scale of the satellite retrievals."*

Fourth, the authors should point out that in this study, they are calibrating here against very accurate model results. Here, they are applying the algorithm in this study over a domain where (in my opinion) the most accurate model results are available anywhere in the world. There is no other mountain range, to my knowledge, with the density of observations available in the Alps. Further, globally available model results in mountain ranges are inadequate for most applications, in terms of their spatial resolution and accuracy. See e.g. Mortimer et al. 2020. I think this needs to be mentioned in the conclusions.

*We agree with the reviewer that the Alps is a region for which very detailed and accurate model simulations are available, which would not be available in most other regions. However, we believe that the availability of distributed snow depth reference data is not critical for the application. Figure 8 shows that the use of distributed snow depth information in the calibration only slightly improves the results, compared to the use of a simple, constant scaling. Please also refer to our response above.*

Fifth, the authors need to acknowledge explicitly that the first four points mean that you could not use this approach globally, calibrated to models, and achieve the kind of results shown here; this point almost certainly will be lost on readers of the abstract alone. This is a major issue with the manuscript that needs to be addressed in the abstract and conclusions.

*The approach with constant scaling factor is applicable globally (in regions with sufficient snow accumulation) and has previously been applied over all mountain chains in the Northern Hemisphere (Lievens et al., 2019). We here show that only a slight reduction in performance is to be expected in the case that insufficient or inaccurate reference data would preclude a (spatial) refinement of the scaling coefficient.*

I hope the authors do not misinterpret any of these comments: they have done an amazing job uncovering this important new dataset. It has very important possible applications. Reworking the way the paper is presented should help the community get on board with this new dataset as quickly as possible.

*Thank you for this supportive comment.*

Mortimer, C., Mudryk, L., Derksen, C., Luojus, K., Brown, R., Kelly, R., & Tedesco, M. (2020). Evaluation of long-term Northern Hemisphere snow water equivalent products. The Cryosphere, 14(5), 1579–1594. https://doi.org/10.5194/tc-14-1579-2020

*Lievens, H., Demuzere, M., Marshall, H.-P., Reichle, R. H., Brucker, L., Branger, I., de Rosnay, P., Dumont, M., Girotto, M., Immerzeel, W. W., Jonas, T., Kim, E. J., Koch, I., Marty, C., Saloranta, T., Schöber J., and De Lannoy, G. J. M., Snow depth variability in the Northern Hemisphere mountains observed from space, Nature Communications, 10, 4629, 2019.*

---

## Author Response (AR2)

Author responses are shown in blue. Page numbers and lines refer to the manuscript with tracked changes.

Dear Dr. Lievens:

Thank you for submission of responses to the initial review comments, and the revised manuscript. Both reviewers have provided their assessment of the revised manuscript, and we are in a situation with divergent recommendations. While Reviewer 2 now recommends the revised manuscript for publication, Reviewer 1 has again recommended rejection. Given this situation, I have also closely reviewed the responses to the review comments and the revised manuscript. My assessment is that two issues require further attention before the manuscript can be accepted. These are outlined below. I would like to preface these comments by affirming that both Reviewers acknowledge the level of effort that has gone into this analysis, and that the work is presented is a well-written fashion. What remains is the clarification of a number of points related to better understanding the physical mechanisms driving the C-band radar response to snow depth, including the impact of Sentinel-1 processing decisions, and the impact of shallow snow (depth < 1 m) on the validation results. Because this work has a high degree of potential impact to the snow remote sensing community and beyond, I think it is important that these issues are satisfactorily addressed. As Reviewer 2 stated in their initial review "Reworking the way the paper is presented should help the community get on board with this new dataset as quickly as possible."

Dear editor, we are very grateful for your efforts to assess our responses to the review comments and the revised manuscript. Please find below our responses to your suggestions regarding the two identified issues that require further attention.

1. Reviewer 1 remains unconvinced that snow depth is driving the C-band radar response after the processing steps that are applied to the backscatter data. It is important to consider here that the Reviewer is not skeptical of a potential relationship between the C-band signal and deep alpine snow, rather the concerns remain focused on the impacts of the radar processing, and ensuring that it is indeed the snow depth that is driving the response.

My recommendation is to add an appendix to the manuscript which clearly traces the impact of all the processing steps on transforming the backscatter values from the raw measurements, using the analysis-ready backscatter illustrated in Figure 4 (time series) as the endpoint. A step by step series of figures showing the impact of multi-looking, reprojection, orbit correction, and averaging (weekly temporal; spatial aggregation) would provide the necessary clarity on how the raw radar data are transformed before the empirical retrieval is applied. This may also help address the finding that "…skill improves with coarser scales…", the physics of which are not presently addressed nor explained.

We appreciate this suggestion, which would indeed provide additional support to our processing methodology. We have decided to go a step further by modifying our processing chain in order to specifically address two out of the four reviewer comments on the processing. More specifically, we (i) de-activated the orbit bias correction step, and (ii) modified the retrieval algorithm to (a) only trace the changes in backscatter between successive observations from the same orbit and (b) provide snow depth estimates at the times of the actual satellite acquisitions to provide daily coverage over the available satellite swaths rather than weekly coverage over the entire Alps. We agree with the reviewer that this will allow to potentially better capture the sub-weekly variability originating from processes such as accumulation, sublimation, wind compaction and re-distribution, etc.

Regarding the other two processing comments (on multi-looking and projection), we are strongly opposed to modifying these basic and standard processing steps, which are moreover indispensable. Not applying multi-looking implies a backscatter processing and snow depth retrieval at the original 10-m pixel spacing that becomes computationally infeasible, both in terms of processing time and storage requirements. Not applying a projection onto a consistent coordinate system impedes a time series analysis (and change

detection) and strongly reduces the applicability of the snow depth output. We also expect by no means that either of these steps can impact the main conclusions of our work.

Please note that potential reasons for skill improvements with coarser scales are discussed in the manuscript at Page 19, Line 34 to Page 21, Line 10, at Page 23, Lines 1-6, as well as in the Conclusions (Page 24, Lines 4-6), and that the spatial aggregation is applied to the snow depth retrievals and not to the backscatter data during processing.

2. The implication of Figures 4, 8, and 10 is that the backscatter is responding to snow depth across the full range of reference snow depth values (0 to ~3 m). There is no evidence in panels a, b, and d of Figure 4, that the retrievals when snow depth is < 1 m are any more uncertain than when snow depth is > 1 m. The clear message from Figures 11 and 12, however, is that uncertainty is greater when snow depth is <1 m. At present, the snow depth dependence of the results is not clearly assessed in Section 4:

"Sentinel-1 (S1) backscatter observations, particularly in VH-polarization, correlate well with regional model simulations of snow depth over Austria and Switzerland." There is no mention of snow depth dependence.

"The main uncertainties in the S1 snow depth retrievals are expected to be caused by wet, shallow and occasional snow cover and forest cover." The influence of wet snow and forest cover on the radar signal are clear, but what is it about shallow snow that causes greater uncertainty? Is it just the ground influence?

Also:
-Do Figures 11 and 12 reflect exactly the same data shown in Figure 10? (I assume yes).
-Can you add some additional text to Section 4 which provides an underlying physical explanation for the depth threshold of ~ 1 m in influencing the retrieval skill?

Figures 8 and 10 (Figures 9 and 11 in the revised version) show a similar spread for shallow snow compared to deep snow, but this similar spread will lead to a larger relative error for shallow snow (as shown in Fig. 12 in the original or Fig. 13 in the revised manuscript). Figure 6 (in the revised version) also reveals higher time series correlations between snow depth and backscatter for areas with more snow accumulation. We expect that in shallow snow, the backscatter observations are more sensitive to the conditions of the ground surface. At the same time, areas with shallow snow are typically also more prone to wet snow conditions and to the dis- and reappearance of snow cover, which we expect to be impacting the backscatter signal at C-band.

In this context, the following statements have been added to the revised manuscript:

Page 7, Lines 21-23: "An additional wet snow detection criterion mainly addresses regions where no strong decrease in gamma0 is observed due to a lower sensitivity to snow wetness. This is for instance expected in regions with shallow or patchy snow cover where the soil scattering contribution may dominate"
Page 12, Lines 33-34 "In regions with shallow maximum snow depths (<1 m), gamma0_VV generally remains relatively constant, with changes typically within +-2 dB"
Page 16, Lines 21-22: "The spatial distribution is however similar to that of gamma0_VH, also showing slight decreases in gamma0_CR in regions with shallow snow and forest cover"
Page 17, Lines 3-7: "It shows that the inclusion of zero snow depths results in higher correlations mainly in regions with shallow and occasional snow (e.g., in eastern Austria). Nevertheless, it is important to remark that the S1 gamma0 observations in these regions only show a weak (if any) correspondence with snow depth, because of the weak scattering contributions from shallow snow, frequent wet snow and melt conditions, and the frequent disappearance and re-appearance of snow cover."
Page 24, Lines 7-14: "The main uncertainties in the S1 snow depth retrievals are expected to be caused by wet, shallow and occasional snow cover and forest cover. Wet snow is known to cause a strong decrease in radar backscatter due to signal absorption. Although a wet snow detection algorithm is implemented, undetected wet snow (for instance due to an insufficient decrease in the backscatter) may cause underestimation in the snow depth retrievals. Uncertainties can also be large in regions with shallow and

occasional snow cover, where the backscatter observations can be dominated by scattering contributions from the ground surface, resulting in a weak (or even negative) correlation with snow depth. For shallow snow conditions, backscatter observations at higher frequencies (e.g., X- or Ku-band), or potentially also using InSAR phase changes at lower (e.g., L- or P-band) frequencies, could be more suitable to detect short-term snow depth changes."

Figures 11 and 12 (Figs. 12 and 13 in the revised manuscript) are derived from exactly the same data as Fig. 10 (now Fig. 11). However, the revised Fig. 12 stratifies the performance metrics by the range in snow depth, thus combining metrics from different sites that reach a similar peak value of snow depth. On the other hand, Fig. 13 stratifies by the actual snow depth, and can thus combine metrics from different time steps at the same site or from different sites. This difference in stratification is mentioned in the caption of Fig. 13.

Thanks very much for considering these comments. My hope is that the addition of a data-focused visualization of the processing chain in an appendix will provide convincing evidence, improved traceability, and increased confidence in this analysis.

Chris Derksen

We thank you for the careful assessment of our work, and sincerely hope that our modifications to the revised manuscript and responses to the review comments comply with the requirements, in order to consider our work for publication in the Cryosphere.

Hans Lievens, on behalf of the co-authors

Reviewer #1

I truly understand the amount of work that was put into this study and the amount of time it takes to reprocess such a large volume of data. I also appreciate the effort and the amount of work and detail that went into this manuscript and revisions.

We thank the reviewer for acknowledging the amount of effort that went into our study over the past 2 years.

Nonetheless, I must recommend this paper be rejected for the same reasons as my initial review. With the different levels of processing, it is impossible for me to clearly say that what is detected is snow depth. A lot of researchers have done similar studies since the early stages of C-band SAR data and many times the relationship found with the signal was not with snow depth but often a contribution from the background signal or the heterogeneity of the snowpack but not it's depth, even in alpine environments.

We regret to see that the reviewer repeated the main previous criticism ("*With the different levels of processing, it is impossible for me to clearly say that what is detected is snow depth*") even though we substantially modified the processing of the backscatter data to address part of the reviewer comments, and despite the proven agreement of our snow depth retrievals with in situ data.

Section 3.1.1. provides a literature overview on previous work using C-band backscatter for snow depth (and SWE) retrieval. As indicated in this section, cross-polarized satellite observations over deep snow have to our best knowledge not been investigated in the past, and certainly not with Sentinel-1 data, providing frequent coverage with a fixed observation scenario. In the previous review comments, the reviewer stated that our literature overview did not apply to the current study, because of the generally different conditions being investigated in past studies (shallow snow often in tundra/taiga environments studied with co-polarized backscatter data). We are therefore somewhat surprised by this contradictory statement ("*A lot of researchers have done similar studies since the early stages of C-band SAR data and many times the relationship found with the signal was not with snow depth but often a contribution from the background*

*signal or the heterogeneity of the snowpack but not it's depth, even in alpine environments."*). We invite the reviewer to provide references to studies that are using cross-polarized C-band backscatter over deep alpine snow to support this statement, which we will gladly integrate into the literature overview.

As for smoothing steps in the processing:

1- Multi-looking is a spatial filtering that yes removes speckle but also smoothes out the images spatially and removes a lot of the high local variability of alpine landscapes.

Multi-looking reduces the grid size by combining all pixels within a certain window into one single (average) value. We are determined to keep the multi-looking as part of the processing chain for the important reason that the alternative (estimating snow depth at the original 10 m pixel spacing) is currently infeasible in terms of computation time and storage requirements.

2- The reprojection steps adds even more spatial smoothing which removes even more spatial variability in high topographic terrain. These two steps are very well known and documented issues in alpine environments.

It is impossible to apply time series analysis and to provide meaningful geospatial information without projecting the data onto a consistent coordinate system.

3- The orbit correction is a temporal filtering. It is known that there are diurnal cycles in the snow conditions especially in alpine snow (melt-refreeze for one). The authors mention in their manuscript that equation 1 uses the "temporal" mean and standard deviation.

The orbit correction is again not 'filtering' nor 'smoothing', but a bias correction applied to the first two order moments of the time series, i.e., the mean and the standard deviation. The temporal mean and standard deviation are taken over the complete time series (and not over a shorter window, which would imply smoothing).

However, to address the reviewer's lack of support for the orbit bias correction, we modified the algorithm in two ways: (i) the orbit bias correction is no longer applied in the backscatter processing, and (ii) the change detection algorithm is adjusted by tracing the differences in backscatter only between acquisitions from the same relative orbit (which are typically 6 or 12 days, or multiples of that, apart). The difference in backscatter is then be imposed onto a weighted average snow index corresponding with the previous observation (6 or 12 days ago). The latter is calculated by applying inverse distance weighting over a certain time window to also incorporate the snow index estimates from other orbits. This modification also allows to retrieve snow depth for every time step when a backscatter acquisition is available (to address point 4 below). For more information, please refer to the revised Section 2.2.

Also, the different orbits and viewing geometries will have different travel paths in the snowpack which the authors neglect to mention has a major impact on the signal intensity and thus smoothes out even more the influence of the snowpack properties on the SAR signal.

We agree that the incidence angle impacts the travel path length of the signal through the snow, and that we should have mentioned this impact in the manuscript. We have investigated the impact of the local incidence angle on the sensitivity of backscatter to snow depth, and found that the sensitivity mainly reduces for angles of 60°-70° or higher. In the reprocessed retrievals, we have masked out backscatter observations with a local incidence angle above 70°. Note that a more stringent masking (excluding also lower local incidence angles) would reduce the spatial coverage of the retrievals.

We have modified Fig. 3 (in the revised version) to show the backscatter time series separately for the different orbits. For the sites in this example, similar sensitivities of backscatter to snow depth are observed for the different orbits in cross-polarization. Stronger differences are found in co-polarization. We are

interested to further investigate the impact of the incidence angle on the retrieval approach in future research.

The following statements have been added to the manuscript:

Page 4, Lines 19-20: "Observations with a local incidence angle >70° were excluded to reduce radar shadowing effects."

Page 12, Lines 2-5: "The gamma0 observations in Austria are shown separately for three selected orbits, i.e., descending orbit 95 (D95), ascending orbit 117 (A117) and D168, corresponding with local incidence angles of 26.5°, 61.9° and 17.8°, respectively. The gamma0 observations in Switzerland are shown for D66 (55.8°), A88 (32.9°) and D139 (46.9°).

Page 12, Lines 7-9: "Limited differences are observed between the orbits, which can be explained by the different local incidence angles (also impacting travel path lengths through the snow), azimuth angles, and overpass times (6 am for D, 6 pm for A)."

Page 12, Lines 18-19: "while also larger differences in backscatter are observed between orbits."

4- The retrieved values are weekly values. This smoothes out all the snow and SAR signal variability over the entire week which can be a lot in alpine conditions (several cm to metes of fresh snow, sublimation, wind compaction, blowing snow, etc.).

First, we do not see any issue with producing weekly average snow depth retrievals as derived from weekly average backscatter. Note that we chose a weekly temporal resolution to (i) indeed average out short-term variability and (ii) to reduce the processing time and storage requirements by computing outputs only once a week instead of daily.

However, we acknowledge that weekly average retrievals indeed no longer contain information on sub-weekly variability caused by processes such as those mentioned by the reviewer. Therefore, we modified the algorithm to calculate the snow depth retrievals at the time of the backscatter observations. We now produced output at the daily time step with coverage according to the Sentinel-1 swaths (instead of the entire Alps being consistently covered in the weekly averages) and validated the results with model simulation and in situ measurements at the daily instead of the weekly timescale. All associated figures (i.e., Figs. 5-13) have been updated accordingly.

For these reasons, it is not possible for me to recommend this paper for publication.

We are thankful to the reviewer for the detailed assessment of our work, and for providing several pathways to improve the retrieval algorithm and the quality of the manuscript. However, in our opinion the comments above do not fully justify the final recommendation of the reviewer. Two out of the 4 processing steps that are criticised (multi-looking and georeferencing) are standard practice, are indispensable, and are by no means impacting the time series sensitivity of the data to geophysical parameters, which is at the basis of our change detection method. To the contrary, these are both necessary steps to keep the processing feasible and to provide meaningful geospatial information.

While we are also convinced that the other 2 criticised processing steps (the orbit bias correction and the weekly temporal resolution of the retrievals) were by no means impacting the main conclusions of the work, we have modified the algorithm to comply with the review comments, by (i) excluding the orbit correction and (ii) producing retrievals at the time of the satellite acquisitions. As the reviewer rightly pointed out, the latter can potentially improve the capturing of sub-weekly processes.

With the above-described modifications to the backscatter processing chain and to the retrieval algorithm, and with the responses in this letter, we are hopeful that the reviewer can settle with our proposed methodology. We would find it difficult to conceive if the remaining discordances on standard processing steps (such as multi-looking and projection onto a coordinate grid) would prevent our study from being

published and would thereby also prevent a novel dataset with demonstrated accuracy from being accompanied by peer-reviewed literature support and documentation.

Reviewer #2

The author have responded adequately to my comments.

We would like to express our thanks for taking the time to review our paper and for the constructive and useful feedback.